# The long noncoding RNA ADIPINT regulates human adipocyte metabolism via pyruvate carboxylase

Alastair G. Kerr [1], Zuoneng Wang[2], Na Wang[1], Kelvin H. M. Kwok[1,3], Jutta Jalkanen[1], Alison Ludzki [1], Simon Lecoutre[1], Dominique Langin [4,5], Martin O. Bergo [3], Ingrid Dahlman[1], Carsten Mim[2], Peter Arner [1,6✉] & Hui Gao [3,6✉]

The pleiotropic function of long noncoding RNAs is well recognized, but their direct role in governing metabolic homeostasis is less understood. Here, we describe a human adipocyte-specific lncRNA, ADIPINT, that regulates pyruvate carboxylase, a pivotal enzyme in energy metabolism. We developed an approach, Targeted RNA-protein identification using Orthogonal Organic Phase Separation, which identifies that ADIPINT binds to pyruvate carboxylase and validated the interaction with electron microscopy. ADIPINT knockdown alters the interactome and decreases the abundance and enzymatic activity of pyruvate carboxylase in the mitochondria. Reduced ADIPINT or pyruvate carboxylase expression lowers adipocyte lipid synthesis, breakdown, and lipid content. In human white adipose tissue, ADIPINT expression is increased in obesity and linked to fat cell size, adipose insulin resistance, and pyruvate carboxylase activity. Thus, we identify ADIPINT as a regulator of lipid metabolism in human white adipocytes, which at least in part is mediated through its interaction with pyruvate carboxylase.

[1] Department of Medicine (H7), Karolinska Institutet, Karolinska University Hospital Huddinge, Huddinge 141 86, Sweden. [2] Department of Biomedical Engineering and Health Systems, Royal Technical Institute, Stockholm, Sweden. [3] Department of Biosciences and Nutrition, Karolinska Institutet, Huddinge 141 83, Sweden. [4] Institute of Metabolic and Cardiovascular Diseases (I2MC), Institut National de la Santé et de la Recherche Médicale (Inserm), Université de Toulouse, UPS, UMR1297 Toulouse, France. [5] Department of Biochemistry, Toulouse University Hospitals, CHU Toulouse, Toulouse, France. [6]These authors jointly supervised this work: Peter Arner, Hui Gao. ✉email: peter.arner@ki.se; hui.gao@ki.se

Multiple disturbances in white adipose tissue (WAT) function are well described and include altered lipid metabolism and ensuing insulin resistance[1–3]. The lipid synthesis and breakdown rates in fat cells govern lipid turnover and thereby fat cell size, which in turn is a key regulator of fat mass[4,5]. Lipids make up more than 95% of white adipocyte volume and thus, by gaining a better understanding of factors that influence the size and metabolism of the lipid droplet, key mechanisms involved in determining adipose depot size can be elucidated.

Long noncoding RNAs (lncRNAs) are a class of polymeric molecules that are known to be differentially expressed in WAT during the development of metabolic diseases[6]. Compared to protein-coding genes, lncRNAs display greater cell-type specificity[7]. Thus, identifying the modes of action for dysregulated, cell-specific lncRNA, may provide insights into how metabolic disturbances occur in a cell-dependent manner and pave the way for cell-specific therapeutics. Much of the work on lncRNAs has focused on how they regulate transcription through chromatin modifications, alter the nuclear architecture, and affect mRNA kinetics[8]. In WAT, a few adipocyte-expressed lncRNAs have been shown to regulate the transcription of lipid metabolic genes[6,9,10]. More recently the post-transcriptional role of lncRNAs has been recognized, demonstrating their ability to act as scaffolds for cell signaling mediators[11] or facilitate the phosphorylation of transcription factors[12]. In human brown-like adipocytes, the lncRNA LINC00473 was shown to interact with mitochondrial and lipid droplet proteins, thereby altering the metabolism of the cell[13].

In this study, we identify a human adipocyte-specific lncRNA (CATG00000106343.1), hereafter named ADIPINT for ADIpocyte specific Pyruvate carboxylase INTeracting RNA. ADIPINT acts post-transcriptionally by binding to and regulating the mitochondrial abundance and interactome of pyruvate carboxylase (PC) and as a result its enzymatic activity. Alterations in ADIPINT expression regulate lipid breakdown/synthesis and total triglyceride mass within fat cells, referred to here as fat cell lipid metabolism. We show that perturbations in PC activity mirror the effects of ADIPINT perturbation on adipocyte lipid metabolism and the WAT expression of ADIPINT is closely linked to in vivo metabolic phenotypes and PC activity.

## Results

### Identification of ADIPINT as a potential regulator of human fat cell lipid metabolism

To discover lncRNAs involved in fat cell lipid metabolism, we examined their WAT expression in a cohort of female patients with obesity, before and two and five years after undergoing Roux-en-Y gastric bypass (RYGB) surgery. RYGB surgery was used as a model as it induces marked body weight changes and alterations in fat cell lipid metabolism[5]. The changes in lncRNA expression were further associated with the changes in fat mass and fat cell size over time. Fat cell size reflects the net balance between synthesis and breakdown of triglycerides and influences WAT insulin sensitivity[14]. Clinical characteristics of the cohort are given in Supplementary Data 1 and have been reported previously[15,16]; there was on average a 32% decrease in body weight following surgery and substantial decreases in fat mass and fat cell size. Compared to pre-surgery, 760 and 650 lncRNA were differentially regulated at two and five years, respectively (Fig. 1a and Supplementary Data 2, 3). All lncRNAs that were differentially expressed in the same direction at two and five years post surgery were compared with 67 lncRNAs previously found to be differentially expressed in WAT of insulin-sensitive compared to insulin-resistant women with obesity[6]. Twelve lncRNAs displayed a significant concordant change in

both clinical cohorts (i.e. higher expression before RYGB and in the insulin-resistant state; or lower expression two and five years post-RYGB and in the insulin-sensitive state) (Supplementary Data 4). ADIPINT (CATG00000106343.1) displayed the greatest fold change of the twelve genes in both cohorts and was among the top 5 most downregulated genes at two and five years post-RYGB (Fig. 1a). In addition, the changes in total fat mass and adipocyte cell volume correlated positively with the changes in ADIPINT expression post-RYGB (Fig. 1b). These data suggest that ADIPINT is linked to fat cell size and further study was devoted to this lncRNA.

ADIPINT is intergenic and located on chromosome 9 (Fig. 1c). None of the three annotated transcripts in the ADIPINT loci have protein coding potential according to CPAT, RNACode, phyloCSF, and sORF riboseq coding prediction softwares. ADIPINT is not conserved between phylogenetically distant species at the sequence or genome synteny level. PacBio-seq in isolated mature adipocytes captured five full length ADIPINT transcripts, revealing transcript 2 to be the dominant isoform (Fig. 1c). RNA-seq and CAGE-seq[17] in human adipose-derived stem cells (hADSC) from subcutaneous WAT supported this transcript-specific expression (Fig. 1c). Transcript 2 is 4.8 kb long and contains two exons (Fig. 1c). Upregulation of ADIPINT expression during in vitro differentiation of hADSC and enrichment in mature adipocytes isolated from WAT were confirmed by qRT-PCR (Fig. 1d, e). RNA expression analyses of different human cell types in the FANTOM5 database[7] revealed that ADIPINT is exclusively expressed in adipocytes (Fig. 1f). Subcellular fractionation studies and RNA-fluorescence in situ hybridization (FISH) revealed ADIPINT to be predominantly localized in the cytoplasm of adipocytes (Fig. 1g, h). The ADIPINT-FISH signal was validated by siRNA-mediated knockdown (Supplementary Fig. 1). Thus, ADIPINT is an adipocyte-specific cytoplasmic enriched lncRNA linked to insulin resistance, fat mass, and adipocyte size.

### ADIPINT expression regulates fat cell lipid metabolism

The influence of ADIPINT on adipocyte function was assessed by knockdown experiments in differentiated hADSC. We used three different anti-sense oligonucleotides (GapmeRs) and found that they reduced ADIPINT expression by 89–97% compared to a negative control GapmeR (GapmeR NC) (Fig. 2a). ADIPINT knockdown reduced basal glycerol release (lipolysis index) by 27–58% (Fig. 2b) but did not alter catecholamine (isoprenaline) stimulated glycerol release (Fig. 2c). Insulin-stimulated lipid synthesis was reduced by 54–67% (Fig. 2d) and total intracellular triglyceride content by 40–66% (Fig. 2e). Consistent with these findings siRNA targeting ADIPINT reduced glycerol release, intracellular triglyceride content, and insulin-stimulated lipid synthesis to the same degree as the GapmeRs, while stimulated lipolysis was unaffected (Supplementary Fig. 2a–e). There were no differences in cell viability after ADIPINT knockdown as assessed by Alamar blue fluorescence (Supplementary Fig. 2f). As lipid metabolism is closely linked to mitochondrial oxidative metabolism[18,19], we therefore examined if the effects of ADIPINT knockdown also influence fat cell energy metabolism. Indeed, ADIPINT knockdown in adipocytes reduced the basal oxygen consumption rate (OCR) (Fig. 2e, i), the glycolytic capacity (Fig. 2f, i) determined after oligomycin treatment, and the extracellular acidification rate (ECAR) (Fig. 2g, j). Therefore, ADIPINT knockdown affects both the aerobic and anaerobic respiration rates and impairs adipocyte lipid metabolism.

To identify the molecular action of ADIPINT, we first carried out transcriptomic and proteomic analyses after knockdown with each of the three ADIPINT-targeting GapmeRs. The downregulated

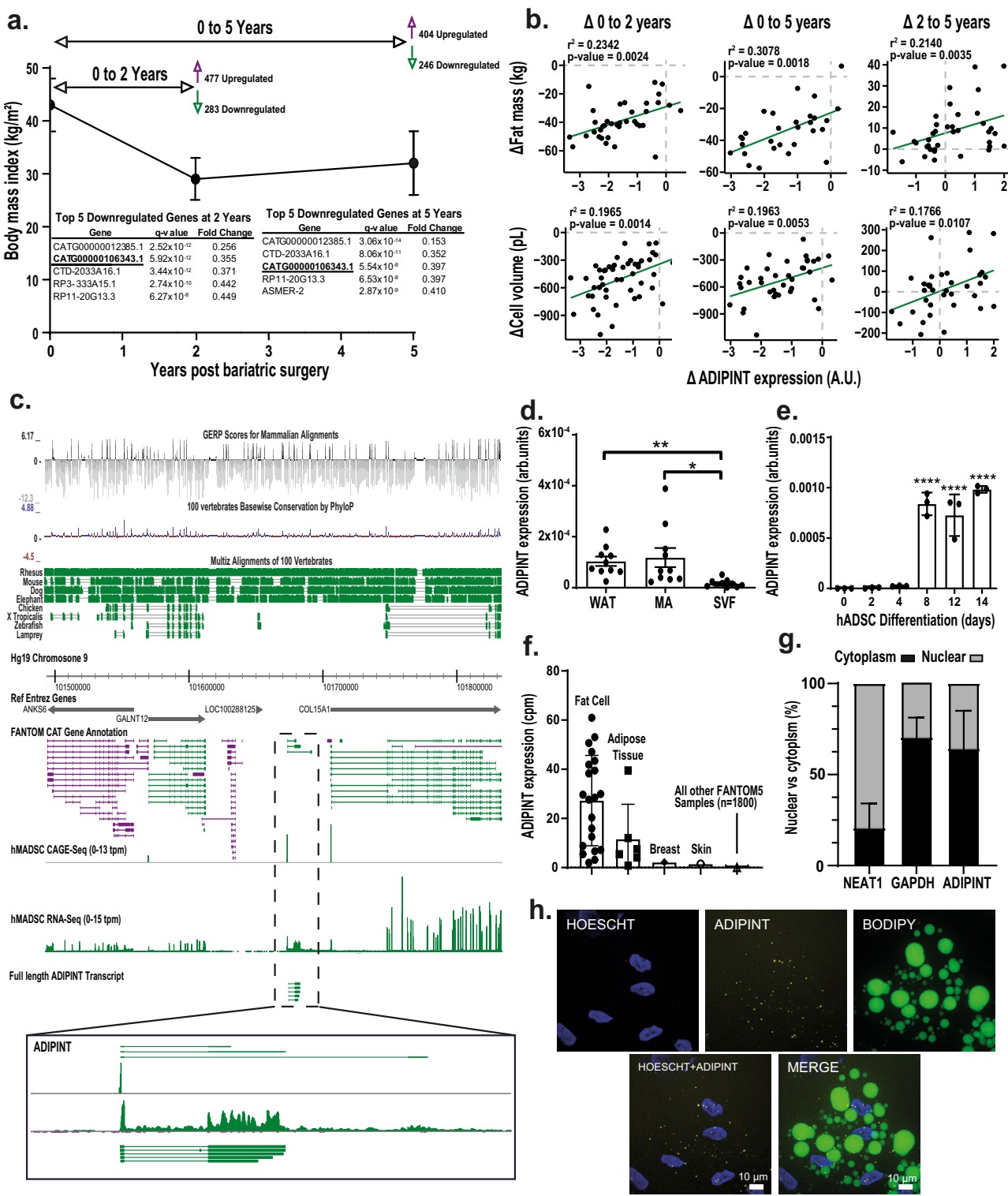

mRNAs were enriched for metabolic pathways including pyruvate metabolism and fatty acid metabolism (Fig. 2k, Supplementary Data 5–8). We did not see changes in *PLIN1*, *ADIPOQ* or *FABP4* expression, indicating it was not a uniform decrease in all adipocyte marker genes following ADIPINT knockdown (Supplementary Data 5). The median fold-changes induced by the three GapmeRs at the RNA and protein levels correlated significantly ($r^2 = 0.2284$, *p*-value < 0.0001; Supplementary Data Fig. 2g). The proteome analysis revealed 10 proteins to be up- and 28 downregulated by each of the three GapmeRs (Supplementary Data 9). Interestingly, 19

of the 38 dysregulated proteins (marked by an asterisk in Fig. 2l) were not significantly regulated at the transcript level, indicating post-transcriptional regulation. Pathway analysis of the 38 dysregulated proteins showed overrepresentation for the GO terms mitochondrial matrix and fatty acid metabolic process (Supplementary Data 10–11) in line with the changes observed above in lipid and energy metabolism. Based on the information at www.genecards.org, we visualized all 38 proteins affected after ADIPINT knockdown in Fig. 2k. Most proteins were downregulated and involved in fatty acid synthesis and oxidation. Proteins within the pyruvate dehydrogenase

**Fig. 1 The adipocyte-specific lncRNA ADIPINT (CATG00000106343.1) is linked to changes in fat mass and fat cell volume. a** Identification of differentially-expressed lncRNAs in white adipose tissue (WAT) two ($n = 49$) and five ($n = 38$) years post-Roux-en-Y gastric bypass (RYGB) surgery compared to pre-RYGB ($n = 50$). The five lncRNA with the greatest decrease in fold change at two and five years are listed, including ADIPINT (CATG00000106343.1). **b** Correlation of the change in ADIPINT expression versus the change in total fat mass or adipocyte cell volume from zero to two, zero to five and two to five years post-RYGB. Each data point represents an individual patient. Linear regression was used to assess significance with no adjustment for multiple comparisons. **c** Genomic position of ADIPINT on chromosome 9 is indicated and the expression detected by RNA-Seq and CAGE-Seq in human adipose-derived stem cells (hADSC) at day 14 of differentiation and the full length transcript determined by PacBio-Seq is shown. The conservation of the ADIPINT genomic region in mammalian and non-mammalian species is displayed above. **d** qRT-PCR analysis of ADIPINT expression in fractionated mature adipocytes (MA) and stroma-vascular fraction cells (SVF) from human WAT, $n = 10$ independent samples. **e** qRT-PCR analysis of ADIPINT during hADSC differentiation into adipocytes, $n = 3$ independent experiments. One-way ANOVA with Dunnett's post-hoc test was used to assess significance compared to day 0 of differentiation and one-way ANOVA with Holm-Sidak's post-hoc test was used to assess significance between WAT fractionation samples. **f** Tissue distribution of ADIPINT expression extracted from the FANTOM5 database. Samples with detected expression (counts per million (cpm)) are plotted, 1829 samples were investigated. **g** Expression of ADIPINT ($n = 3$) in fractionated hADSC. NEAT1 ($n = 6$) and GAPDH ($n = 6$) were used to validate the nuclear and cytoplasmic fractions, respectively. **h** RNA-FISH for ADIPINT in differentiated hADSC, staining for nuclei (HOECHST-Blue) and lipid (BODIPY-Green) alongside fluorescent probes targeting ADIPINT (Yellow). The experiment was performed three times with similar results. Scale bar, 10 μm. Data are presented as mean values +/- SD for $n < 6$ and SEM for $n \geq 6$. *$p < 0.05$, **$p < 0.01$, ****$p < 0.0001$. Source data are provided as a Source Data file.

complex were downregulated, while pyruvate kinase isozymes R/L, involved in synthesizing pyruvate, were upregulated. Together these data indicate a clear metabolic effect of ADIPINT on lipid and energy metabolism, including the regulation of key components in the mitochondria.

**ADIPINT interacts with pyruvate carboxylase**. To understand the mechanism by which ADIPINT regulates adipocyte function, we identified its direct protein interacting partners. Recently, orthogonal organic phase separation (OOPS)[20] enabled the non-targeted isolation of the entire RNA-bound proteome. We applied OOPS to enrich the RNA-bound proteome and then performed targeted pulldown of ADIPINT in human white adipocytes. We named this method Targeted RNA-protein identification using OOPS (TROOPS, see Methods for details). Adipocytes underwent UV crosslinking, followed by Trizol-chloroform extraction, where RNA-protein complexes migrate to the interphase and unbound RNA, protein, and DNA in other fractions can be removed (Fig. 3a). The interphase is cleaned, sonicated and used for targeted pulldown of ADIPINT with short antisense oligonucleotides. After UV crosslinking, ADIPINT became enriched at the interphase, indicating its binding to a protein partner (Supplementary Data Fig. 3a). Using two distinct sets of oligonucleotides, each tiled along ADIPINT, we enriched the lncRNA 3.5–4.8 fold compared to the input material (Fig. 3b). No ADIPINT expression was detected after pulldown using probes targeting LacZ mRNA or using the ADIPINT-targeting probes with RNase. Mass-spectrometry analyses revealed that pyruvate carboxylase (PC) was the only significantly enriched protein following ADIPINT pulldown (Fig. 3c). No enrichment in PC was found in the RNase-treated or LacZ probe sets. All detected proteins in each experiment are listed in Supplementary Data 12. We confirmed the interaction with PC by western blot analysis (Fig. 3d). GAPDH, another enzyme known to bind RNA[21], was used as a control for pulldown and interaction specificity. To validate the ADIPINT interaction, we performed immunoprecipitation of PC in hADSC. ADIPINT was enriched when compared to IgG and this enrichment was significantly higher than that of 18S, PC, and U1 RNA (Fig. 3e). The ADIPINT-PC interaction was further assessed by size exclusion chromatography (SEC). Both the RNA and protein alone could be separated using a Superose® 6 Increase 10/300 GL column (Supplementary Data Fig. 3b). When ADIPINT and PC were combined and the resulting mixture loaded, PC and ADIPINT eluted together in the RNA elution fraction (Fig. 3f, Supplementary Data Fig. 3b). PC when run alone or with ADIPINT anti-sense did not elute at the RNA elution volume to

any significant degree (Fig. 3f). To visualize a direct interaction between ADIPINT and PC we performed electron microscopy after immuno-gold labelling of PC and detected multiple PC molecules on the surface of ADIPINT after purification of the complex (Fig. 3g, Supplementary Data Fig. 3c). We confirmed this stain to be specific for PC and did not detect a PC antibody signal localized to ADIPINT anti-sense (Supplementary Data Fig. 3d–h). Dual immuno-gold staining against PC and digoxin-labelled ADIPINT confirmed both components in the complex (Fig. 3h, Supplementary Data Fig. 3i, j). The particles visible in the electron micrographs were both reactive to the 10 nm gold-labelled digoxin antibody (Protein A, red arrows) and the 20 nm gold labelled PC antibody (white arrows). We next examined if PC interacts with a specific region of the ADIPINT transcript. ADIPINT was divided in two, each transcript representing one half of the full-length RNA. SEC purification of the truncated ADIPINT-PC complexes showed that the 5′ half maintained the affinity to bind PC (Fig. 3i). The electron microscopy images confirmed that the first half of ADIPINT has a higher affinity for PC (Fig. 3j). The 3′ half RNA showed a steep drop in gold particles on the ADIPINT molecules (Fig. 3k). Quantification of gold particles localized to the respective ADIPINT RNAs was higher in the 5′ half RNA than the 3′ half (Fig. 3l). Furthermore, we found ADIPINT and PC to exist in the same compartments within the cell after performing subcellular fractionation for mitochondrial and cytoplasmic fractions (Supplementary Data Fig. 3k–m). Together these data suggest a physical ADIPINT-PC interaction.

**ADIPINT knockdown reduces pyruvate carboxylase activity by altering its localization and interactome**. The enzymatic activity of PC increases during adipogenesis[19], which leads to increased production of oxaloacetate used in fatty-acid, glycerol, and trigly-ceride synthesis[22]. PC protein expression was not altered after ADIPINT knockdown (Supplementary Data 9), and we therefore hypothesized that the ADIPINT-PC interaction might directly regulate PC function. ADIPINT was knocked down using three separate GapmeRs (1–3) and PC activity was measured as the rate of conversion of pyruvate to oxaloacetate (Supplementary Data Fig. 4a). The time for reaching 50% completion of the reaction was 1.6–2.7 fold greater for GapmeRs 1–3, compared to GapmeR NC (Supplementary Data Fig. 4b). The initial velocity after ADIPINT knockdown was 19.9–33.1% lower than the control GapmeR (Fig. 4a). No change was seen in PC protein expression in the same experiments (Fig. 4b). Normalising the enzyme activity by protein expression, we found that PC's activity was reduced upon ADIPINT knockdown (Fig. 4c). The decrease in PC activity was

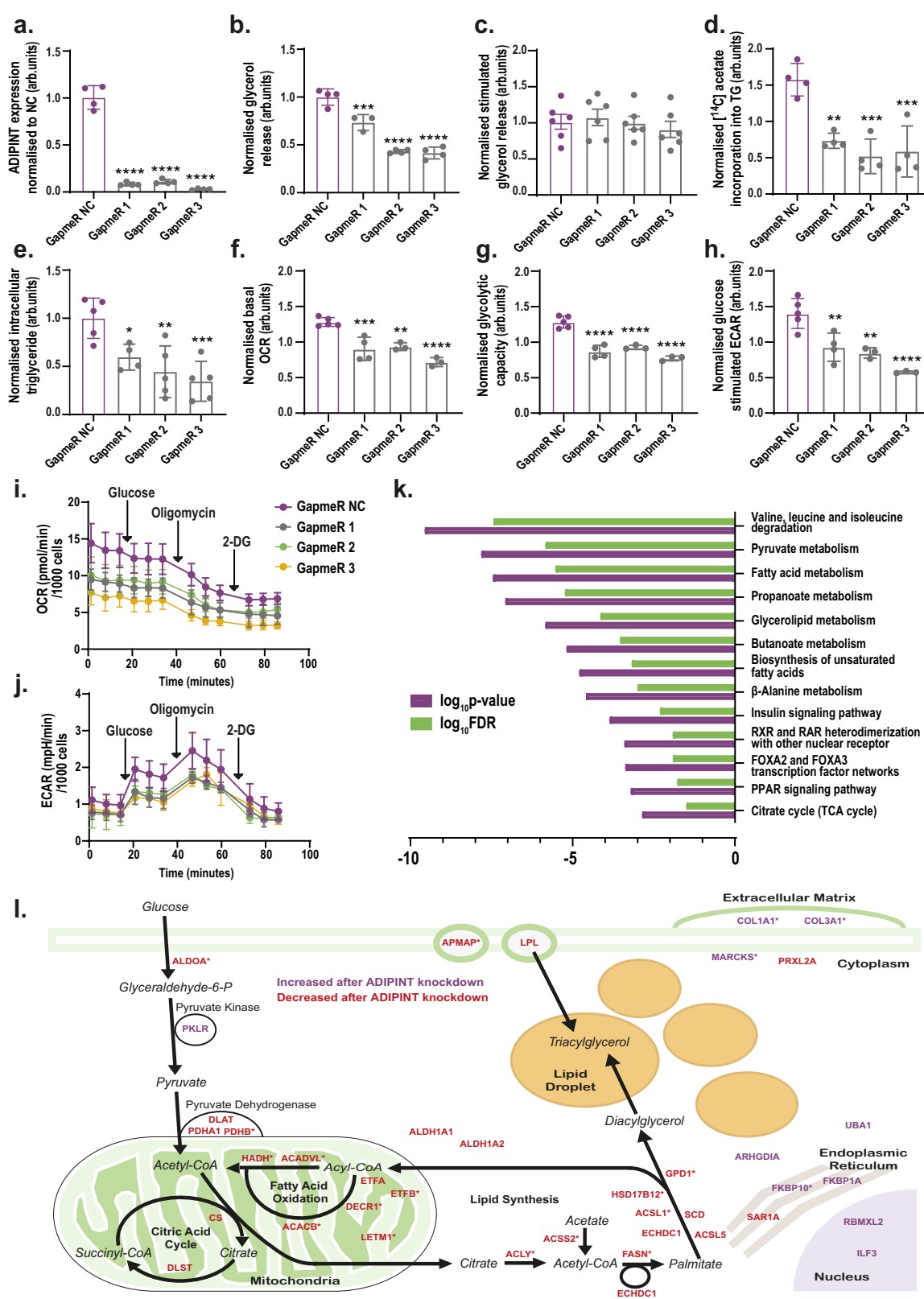

accompanied by reduced levels of intracellular oxaloacetate (Fig. 4d); intracellular pyruvate levels were unaffected (Fig. 4e). Thus, lower oxaloacetate concentration was not the result of lower pyruvate concentration. As PC produces oxaloacetate in the mitochondrial matrix we determined if the changes in oxaloacetate was related to less PC in the mitochondria. Consistent with previous data, western blots of whole-cell lysates revealed that the total

amount of PC within the cell did not change following ADIPINT knockdown (Fig. 4f, g). However, the amount of PC within the mitochondrial/membrane fraction decreased after knockdown (Fig. 4f, h) and the activity of PC within this fraction was also reduced (Fig. 4i, Supplementary Data Fig. 4c).

Metabolic enzymes are compartmentalized in a cell putting them in close proximity of sequential enzymes within the

**Fig. 2 Knockdown of ADIPINT perturbs lipid metabolism and bioenergetics. a** ADIPINT expression ($n = 4$) **b** glycerol release (measure of basal lipolysis) ($n = 3,4$) **c** isoprenaline stimulated glycerol release (measure of stimulated lipolysis) ($n = 6$) **d** insulin stimulated [$^{14}$C] acetate incorporation into intracellular lipid (measure of lipid synthesis) ($n = 4$) **e** intracellular triglyceride (TG) content ($n = 4,5$) **f** basal oxygen consumption rate (OCR) ($n = 3$-5) **g** glycolytic capacity ($n = 3$-5) and **h** glucose-stimulated extracellular acidification rate (ECAR) ($n = 3$-5) in differentiated hADSC transfected with three ADIPINT targeting GapmeRs (1–3) or non-targeting control GapmeR (NC). Each data point represents an individual experiment. Data was normalized to the mean of each replicate experiment apart from **d** mean normalized to GapmeR NC (-insulin). ATP linked OCR was determined after oligomycin administration. ECAR determined after glucose treatment. **i** The OCR and **j** ECAR measured at basal and then after glucose, oligomycin and 2-deoxyglucose (2-DG) treatment over the time course as indicated, $n = 5$ for GapmeR NC, $n = 4$ for GapmeR 1 and $n = 3$ for GapmeR 2 and 3 independent experiments. **a**–**h** Each data point represents an independent experiment. One-way ANOVA with Dunnett's post-hoc test was used to assess significance between GapmeR NC and GapmeRs 1–3. **k** Pathways enriched by genes downregulated after ADIPINT knockdown by each GapmeR independently, compared to GapmeR NC. **l** A schematic diagram of all 38 proteins significantly altered independently by all three GapmeRs targeting ADIPINT. Proteins are displayed based on the associated metabolic pathway and localization according to www.genecards.org. Proteins downregulated following ADIPINT knockdown are in red font and upregulated are in purple font. Proteins marked with * were not significantly regulated at the RNA level. If no metabolic pathway was listed for the given protein, then the protein was placed on the diagram based on the given subcellular localization. Metabolites were not measured. Data are presented as mean values $+/-$ SD for $n < 6$ and SEM for $n \geq 6$. *$p < 0.05$, **$p < 0.01$, ***$p < 0.001$, ****$p < 0.0001$. Source data are provided as a Source Data file.

metabolic pathway and the production of their substrate[23]. When enzymes are assembled into these specific metabolic complexes the activity for that particular enzyme is also enhanced[24]. Consequently, we asked whether ADIPINT might be important for maintaining the PC interactome. We immunoprecipitated PC after ADIPINT knockdown and analyzed the changes in protein partners in the mitochondria/membrane and cytoplasmic fractions using mass-spectrometry. The amount of PC immunoprecipitated from each fraction in GapmeR 1- and GapmeR NC-treated cells was similar, so comparisons of interacting protein abundance could be made (Supplementary Data Fig. 4d, e). As with changes in PC localization and activity, the enriched proteins displayed greater differences after ADIPINT knockdown in the mitochondria/membrane fraction compared with the cytosolic (Supplementary Data Fig. 4f). We therefore focused our analyses of PC-interacting proteins on the mitochondria/membrane fraction. 19 proteins were significantly reduced by eight fold or more in PC immunoprecipitates after ADIPINT knockdown (Supplementary Data 13). These proteins were enriched for the Glycolysis/Gluconeogenesis pathway (Fig. 4j). In adipocytes, gluconeogenesis does not occur due to a lack of the enzyme glucose-6-phosphatase, instead pyruvate is converted to glycerol used for triglyceride synthesis, known as glyceroneogenesis[25]. Examining all the proteins in the Glycolysis/Gluconeogenesis pathway, we found that the subset involved in glycerol synthesis was reduced in PC immunoprecipitates after ADIPINT knockdown, including the rate-limiting enzyme within this pathway, PCK1[26] (Fig. 4k). ADIPINT thus appears to establish a unique interactome and localization of PC, which we suggest may regulate the activity and cellular function of the enzyme.

**Inhibiting pyruvate carboxylase perturbs lipid and energy metabolism in fat cells.** Inhibiting PC activity in mouse adipocytes reduces lipid synthesis and triglyceride accumulation[27]. However, the role of PC in human fat cells still needs to be investigated. To determine if a reduction in PC activity mediates the alterations in lipid metabolism seen with ADIPINT knockdown, we incubated hADSCs with the PC inhibitor oxalate[28,29]. Oxalate treatment reduced glycerol release, intracellular triglyceride, and insulin-stimulated lipid synthesis (Fig. 5a–c). At 500 μM of oxalate, the reduction of lipid synthesis, breakdown, and accumulation was similar to ADIPINT knockdown in parallel experiments using GapmeR 1. We next assessed the effect of PC inhibition on adipocyte bioenergetics. Oxalate administration reduced basal and ATP-linked OCR as well as glucose-stimulated ECAR to a similar degree as was seen following ADIPINT

knockdown (Supplementary Data Fig. 5a–e). To confirm the findings with oxalate treatment, we investigated a second inhibitor of PC activity, avidin, using previously described concentrations[30]. At 0.01 μM avidin, there was a significant reduction in glycerol release, intracellular triglyceride content, and lipid synthesis (Fig. 5d–f). As a third validation, PC was knocked down using an siRNA (Fig. 5g, h). Glycerol release, intracellular triglyceride, and lipid synthesis were all lowered upon knockdown (Fig. 5i–k). Thus, pharmacologic and genetic PC inhibition reduces lipid and energy metabolism to a similar extent as ADIPINT knockdown. To examine if there was an ADIPINT effect beyond PC, we investigated their dual effect on the total triglyceride level (Fig. 5l). After PC inhibition with 0.1 μM Avidin, there was no additional decrease with ADIPINT knockdown. Furthermore, combined PC knockdown and avidin administration did not further decrease triglycerides, suggesting that 0.1 μM avidin resulted in the maximal reduction in triglycerides possible through PC inhibition. Together these data suggest that the ADIPINT-PC axis can regulate the total triglyceride content of the adipocyte.

**ADIPINT expression correlates with adipose pyruvate carboxylase activity and clinical phenotypes.** Finally, we examined the possible physiological impact of ADIPINT. A correlation between ADIPINT and PC was examined in abdominal subcutaneous adipose tissue of 10 lean and 9 with obesity but otherwise apparently healthy women (Cohort 2, clinical data are given in Supplementary Data 14). ADIPINT expression was increased in obesity (Fig. 6a). PC protein expression did not differ between groups (Fig. 6b, Supplementary Data Fig. 6a). However, PC activity was higher in women with obesity (Fig. 6c). When combining lean and obese in Cohort 2, ADIPINT expression and PC activity were positively correlated (Fig. 6d) while no correlation between PC protein expression and ADIPINT was observed (Supplementary Data Fig. 5b). We next examined a larger cohort of 20 women with and 20 without obesity (Cohort 3). Again, ADIPINT expression was increased in women with obesity (Fig. 6e). To determine whether ADIPINT is associated with other clinical traits, the women in cohorts 2 and 3 were combined. ADIPINT expression correlated positively with body mass index, body fat, fat cell volume, and plasma triglyceride levels (Fig. 6f–h, Supplementary Data Fig. 6c); and with overall levels of in vivo insulin resistance (measured by the homeostasis assessment method, HOMO-IR) and in vivo adipose insulin resistance (ADIPO-IR) (Fig. 6i, j). The strongest correlation was between ADIPINT expression and fat cell volume. Almost 60% of the variation in fat cell volume between subjects could be explained

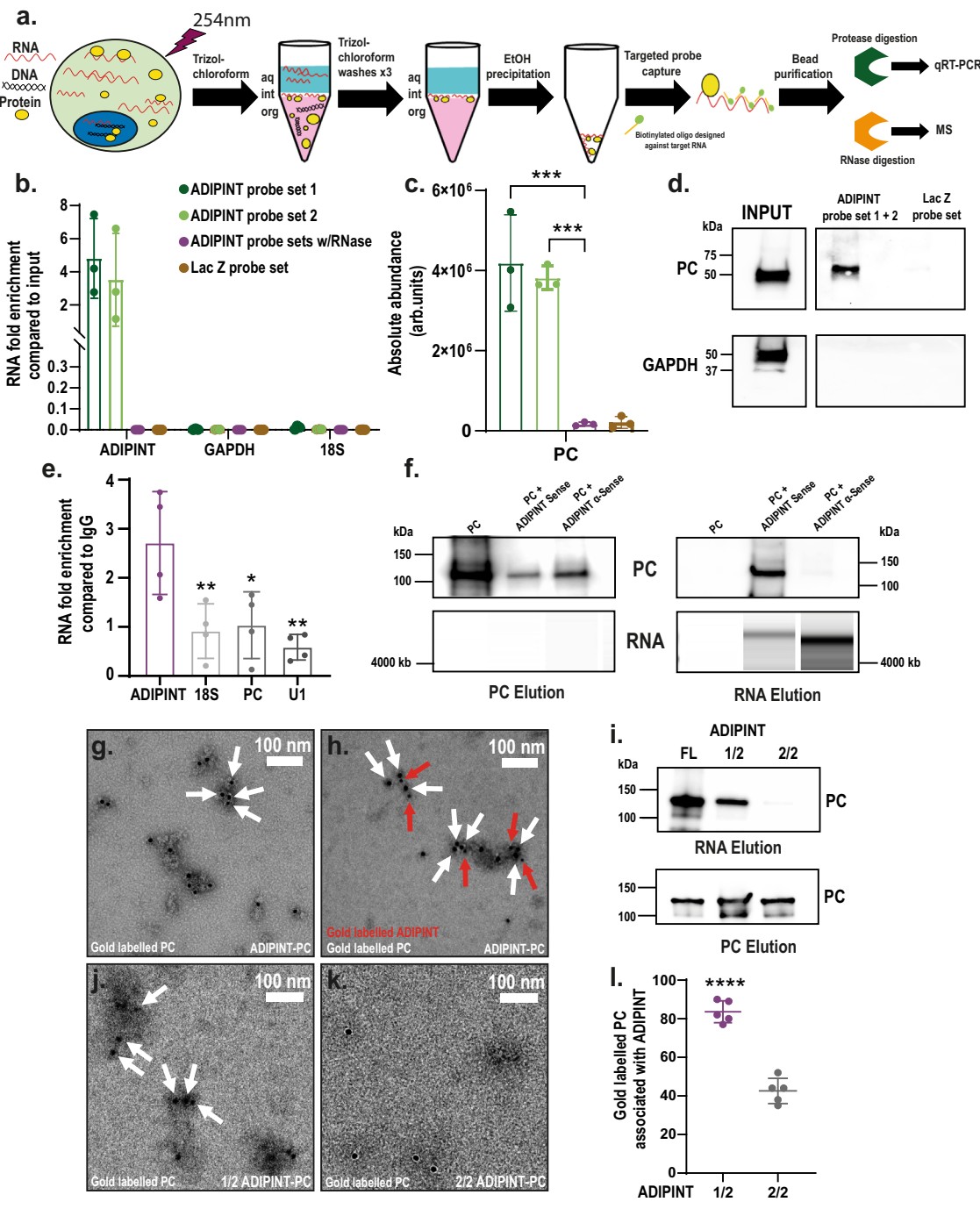

by variations in ADIPINT expression (i.e. adjusted $r^2$). It is well known that body mass index and body fat also strongly influence fat cell volume, however, multiple regression analyses indicate the relationship between ADIPINT expression and fat cell volume is the only independent regressor in this model (Supplementary Data 15).

## Discussion

We propose that ADIPINT functions to enhance the synthesis and breakdown of triglycerides in human fat cells (Fig. 7). The formation and hydrolysis of triglycerides regulate their net storage and in turn the fat cell size. Using two independent strategies, we demonstrate that ADIPINT knockdown reduces the lipid content in fat cells. Therefore, our data suggest that ADIPINT

expression, at least to some degree, can determine the size of the lipid droplet/fat cell. As the ADIPINT actions on synthesis and breakdown of triglycerides have opposite effects on the net lipid content, it appears that the synthesis action predominates. However, we cannot exclude other effects, not examined herein, such as on fatty acid re-esterification and lipogenesis. ADIPINT may exert additional effects on fat cells in relation to energy metabolism as judged by the aerobic and anaerobic respiration rate decreases following knockdown.

Our data suggest that ADIPINT acts by mediating the function of the enzyme PC. Using three different methods (TROOPS, RIP and electron microscopy) we demonstrate that ADIPINT physically interacts with PC. No other significant protein interaction was detected and so we conclude that PC may be the most common protein interactor of ADIPINT. Knockdown or

**Fig. 3 Targeted RNA-protein identification using orthogonal organic phase separation (TROOPS) identifies pyruvate carboxylase as an interacting partner of ADIPINT. a** Graphical representation of TROOPS methodology. **b** qRT-PCR analysis of ADIPINT, 18 S and GAPDH expression in TROOPS pulldown samples. **c** Abundance values obtained from mass-spectrometry for pyruvate carboxylase for probe sets targeting ADIPINT or Lac Z. $n = 3$ independent experiments, statistical comparisons were made between each probe set and ADIPINT probes with RNase using one-way ANOVA with Dunnett's post-hoc test. **d** Western blot analysis of PC and GAPDH after TROOPS pulldown using probes targeting ADIPINT or Lac Z (PC band was validated by mass-spectrometry). **e** RNA immunoprecipitation pulldown using PC or IgG targeting antibody followed by qRT-PCR for ADIPINT, 18 S, PC and U1. The fold increase in PC pulldown samples compared to IgG is plotted and statistical comparisons were made using one-way ANOVA with Dunnett's post-hoc test, $n = 4$ independent experiments. **f** Western blot analysis (top panel) for PC and BioAnalyzer RNA analysis (bottom panel) after SEC of PC alone or in the prescence of ADIPINT or ADIPINT anti-sense. The fractions that PC and ADIPINT/ADIPINT anti-sense elute at were collected (labelled PC elution and RNA elution) and analysed. **g** Representative negative stain electron micrograph of ADIPINT-PC complexes labeled with anti-PC immuno-gold (20 nm). **h** Representative negative stain electron micrograph of ADIPINT-PC complexes double labeled with gold conjugated anti-PC (20 nm, white arrows) and anti-digoxin (10 nm, red arrows). **i** Western blot analysis for PC in the RNA elution and PC elution fraction after SEC using the 5' half of ADIPINT (1/2) or 3' half of ADIPINT (2/2), the full length ADIPINT was used as a control for PC binding. Representative negative stain electron micrograph of **j** 1/2 ADIPINT-PC or **k** 2/2 ADIPINT-PC following SEC. White arrows indicate 20 nm gold particles. **l** Counting of gold labelled particles associated with either 1/2 ADIPINT or 2/2 ADIPINT, $n = 5$ separate images. Unpaired two-tailed $t$-test was used to assess significance. Scale bar is 100 nm for micrographs. Data are presented as mean values +/- SD. *$p < 0.05$, **$p < 0.01$, ***$p < 0.001$, ****$p < 0.0001$. Source data are provided as a Source Data file.

enzymatic inhibition of PC with two different inhibitors displayed similar effects on lipid synthesis, lipolysis, and triglyceride content as ADIPINT knockdown. In addition, PC inhibition showed similar effects on energy metabolism as ADIPINT knockdown. Knockdown of ADIPINT in the presence of PC inhibition resulted in no further reductions in fat cell lipid content seen with PC inhibition alone, suggesting that ADIPINT acts through the same pathway as PC in controlling lipid content. PC is involved in the production of fatty acids and glycerol within adipocytes[19,25] and we show that this regulates the total intracellular triglyceride stores in vitro.

The recent discovery that many enzymes can bind RNA has added a new layer of metabolic regulation[31]. In theory, ADIPINT could influence PC in many ways. Our data suggest that ADIPINT's major action is to function as a mitochondrial gatekeeper for PC, allowing this enzyme to exert its multiple effects on fat cell metabolism within this cellular fraction (Fig. 7). The evidence for this idea is, first, that ADIPIINT knockdown selectively reduced PC abundance in mitochondria and, second, that the knockdown markedly altered the PC interactome within the isolated mitochondria/membrane fraction. We found that in the presence of ADIPINT, PC interacted with enzymes involved in glyceroneogenesis including the rate limiting enzyme, PCK1[26,32]. Other lncRNA have recently been shown to regulate the function of an enzyme by changing its interactome[33,34]. We also observed small but significant changes in PC activity after ADIPINT knockdown. Enzymes form assemblies with other sequential enzymes within a metabolic pathway termed metabolons[23]. These metabolons allow substrate channeling, increasing the flux through the pathway and altering the reaction rate for the particular enzyme[24]. It is possible the aforementioned small changes in activity reflect the changed interactome of PC rather than a direct modulation of PC activity by ADIPINT. However, it needs to be investigated how the lower expressed ADIPINT can regulate the highly abundant enzyme, PC. This obstacle may demand development of novel methods to precisely determine the stoichiometry between ADIPINT and PC in specific subcellular compartments. A similar relationship between a lncRNA and interacting metabolic enzymes was recently seen in breast cancer cells[35]. As PC is present in many cell types and tissues, it will be interesting to understand if the adipocyte-specific RNA ADIPINT is able to establish an adipocyte specific role for PC, linking PC to a particular function in WAT.

What is the physiological role of ADIPINT in vivo? We can only speculate as the gene is human specific and the relevance of studying a human lncRNA in mouse is unclear. Furthermore, we only examined women, so ADIPINT's role in males is presently

unknown. However, our data from three different female cohorts indicate that ADIPINT is important for the body fat level. Indeed, ADIPINT expression in WAT consistently decreased following weight loss and increased in women with obesity. It is unlikely that the findings in Cohort 1 are secondary to the caloric state of the women, as decreased ADIPINT expression was observed two years after RYGB when the patients were still in the catabolic state and also after five years when they were in an anabolic state (as seen with changes in body mass index throughout the study). Furthermore, ADIPINT expression correlated positively with body mass index and body fat levels in many women who were body weight stable at the time of investigation. It is possible that ADIPINT plays some role in metabolic regulation because its expression correlated positively with two different measures of in vivo insulin sensitivity and with the circulating triglyceride level. However, the strongest correlation was with fat cell size, which was independent of the major cofactors body mass index and body fat content. These data in combination with the in vitro findings with fat cell lipid content make it tempting to suggest that ADIPINT is involved in the regulation of lipid droplet/fat cell size. Clearly this speculation must be confirmed by additional studies such as examining ADIPINT expression in persons with equal body fat levels and fat cell size. Finally, we present indirect evidence for a physiological role of the ADIPINT-PC interaction. Thus, adipose PC activity was upregulated in obesity and correlated positively with ADIPINT expression.

Although the contribution of PC to triglyceride homeostasis in vivo has not been extensively studied, existing evidence supports an important role. Knockdown of PC in rat adipose tissue and liver reduces adiposity and improves insulin sensitivity through decreased lipid re-esterification[36]. It has recently been shown in mice that hepatocyte-specific knockout of PC did not alter body weight or adiposity on chow or high fat diets[37]. These rodent studies suggest that PC is important for the WAT mass and insulin sensitivity. Consequently, the fat cell–specific lncRNA ADIPINT could offer a unique way to alter adipocyte PC activity without affecting non-adipose tissues.

Our further development of methods to study RNA-protein interactions (TROOPS) greatly facilitated the observation that ADIPINT and PC interact. Other methods have previously been developed to investigate the binding of lncRNA to protein[38,39] in vivo, though large numbers of cells per replicate are required. By first isolating the RNA-bound proteome before performing targeted pulldown we have markedly reduced the number of cells required to identify interacting proteins for a given RNA. This modification may be useful in other stem cell or primary cell systems where it is difficult to obtain large numbers of cells.

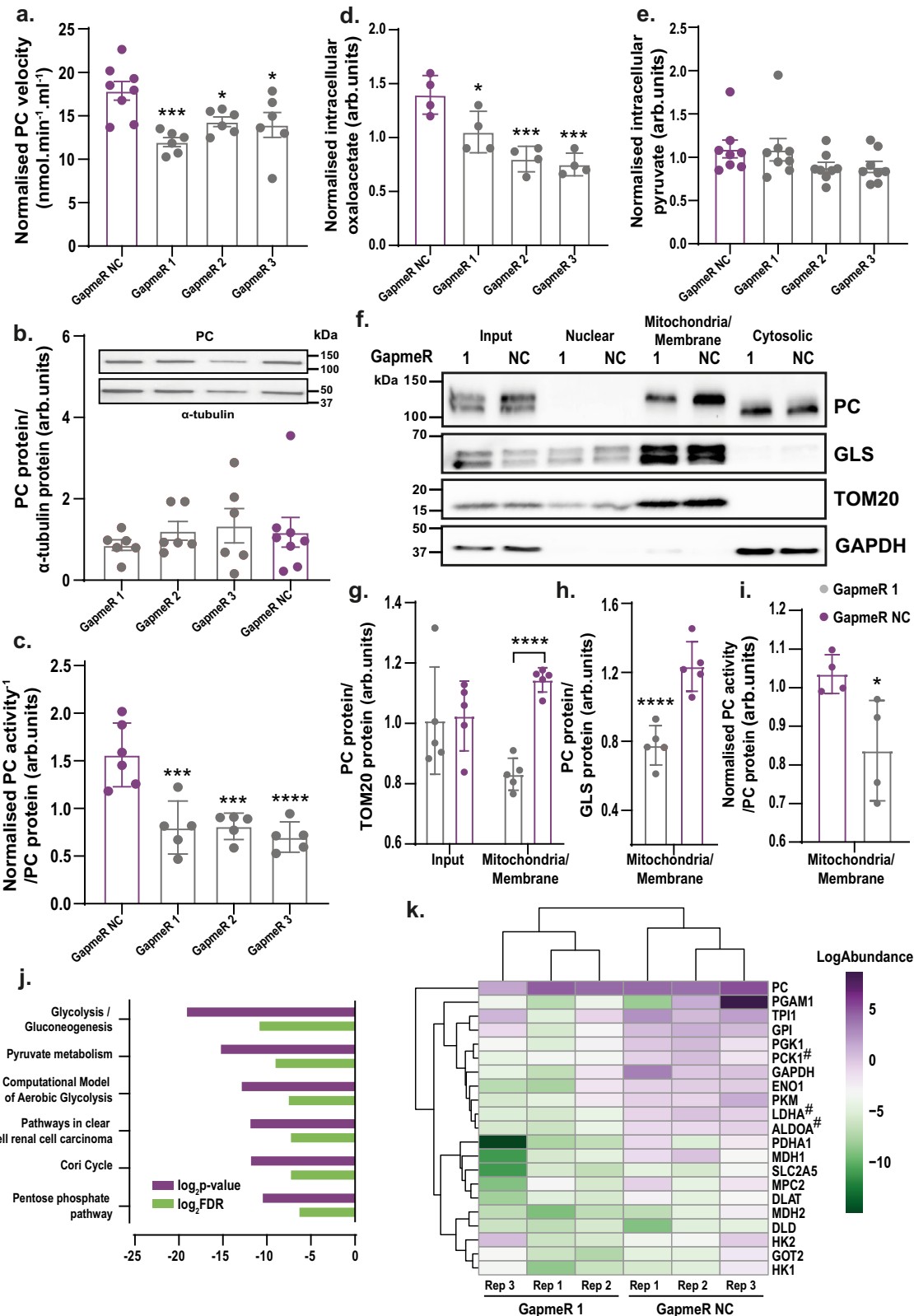

Previously, PC was only detected as an RNA binding protein when using OOPS[20] or orthogonal approaches[40] in one of the five cell lines examined (endothelial), indicating its interaction with RNA may be cell-type specific. Importantly, we were able to validate the interaction identified with TROOPS with several independent methods.

In short, ADIPINT is an adipocyte-specific polynucleotide that regulates lipid metabolism in part by acting as a gatekeeper of PC function. The exact molecular mechanism leading ADIPINT to alter PC remains to be defined. Further studies are needed to establish the physiological consequences of this RNA-protein interaction and determine its suitability as a therapeutic target.

**Fig. 4 ADIPINT knockdown alters mitochondrial pyruvate carboxylase abundance and interactome. a** PC enzymatic velocity measured through a change in absorbance of NADH (340 nm) after transfection with GapmeR NC (black) or ADIPINT-targeting GapmeRs 1–3. Velocity was measured over 20 min ($n = 6$-8). **b** Western blot quantification of PC protein and α-tubulin (housekeeping protein) in the same sample PC activity was measured. PC expression was normalized to α-tubulin. One representative western blot for PC and α-tubulin is given above ($n = 6$–8). **c** PC activity (measured as the time to 50% completion of the activity assay) expressed per amount of PC protein. PC activity and protein were normalized within each experiment and each data point represents an individual experiment ($n = 5$–6). **d** Intracellular oxaloacetate ($n = 4$) and **e** pyruvate levels ($n = 8$) normalized per experiment after ADIPINT knockdown using GapmeRs 1-3. **a–e** Target and control GapmeRs were compared by one-way ANOVA with Dunnett's post-hoc test to assess significance. **f** Representative western blot for PC, the mitochondria-localized proteins TOM20 and Glutaminase (GLS) and the cytoplasmic protein GAPDH after cellular fractionation into nuclei, mitochondrial/membrane and cytoplasmic fractions. **g** Quantification of PC abundance in input and mitochondria/membrane fractions normalized to TOM20 ($n = 5$) and **h** GLS ($n = 5$). **i** PC activity measured in the mitochondria/membrane fraction after ADIPINT knockdown using GapmeR 1 ($n = 4$). Unpaired two-tailed $t$-test was used to compare significance between GapmeR 1 and Gapmer NC samples in the input and mitochondrial fractions. **a–e** and **g–i** each data point represents an independent experiment. **j** PC immunoprecipitation in the mitochondria/membrane fraction after ADIPINT knockdown (GapmeR 1) and compared to control (GapmeR NC) cells. Pathways enriched by proteins significantly altered between the two are conditions are displayed. **k** Proteins detected that map to the wikipathways Glycolysis and Gluconeogenesis (Homo sapiens) pathway are displayed with the log abundance values for each protein and each replicate. Proteins significantly reduced in PC immunoprecipitation samples after ADIPINT knockdown are marked with a #. Data are presented as mean values $+/-$ SD for $n < 6$ and SEM for $n \geq 6$. *$p < 0.05$, **$p < 0.01$, ***$p < 0.001$, #$p < 0.001$, ****$p < 0.0001$. Source data are provided as a Source Data file.

## Methods

**Clinical cohorts**. All studies on human subjects (i.e Cohort 1–3) were approved by the local committee on ethics (Regional etikprövningsnämnden i Stockholm, Karolinska Institutet, Tomtebodavägen 18 A 17165 Solna, Sweden) and explained in detail to each participant who gave informed written consent. No participant was compensated for these studies.

We included three cohorts of women. Cohort 1 had obesity and were subjected to Roux-en-y gastric bypass (RYGB) and the primary outcomes have been reported previously[15,16] (DEOSH, trial registration number NCT01785134). Clinical data extracted from previous publications[16] are presented in Table S1 and Fig. 1a. They were investigated before, 2 and 5 years after RYGB. Three had type-2 diabetes pre-surgery treated with metformin and/or diet only. Diabetes was absent following RYGB. Sixteen women had hypertension at baseline and were treated with ACE- or angiotensin 2 receptor blockers, beta-blockers, thiazide diuretics and/or calcium channel antagonists; 9 women remained on antihypertensive treatment at two years follow up. Two women were on statins pre-surgery; one remained after surgery. All women received dietary instructions following surgery of a protein-rich diet for six weeks. Eating subsequently three small meals rich in protein and whole grains and four small snacks per day was advised. In Cohort 2, we investigated 10 lean women (body mass index <25.0 kg/m²) and 9 women with obesity (body mass index ≥30.0 kg/m²). In Cohort 3, we investigated 19 women without (body mass index <30.0 kg/m²) and 21 with obesity. All women in Cohorts 2–3 were middle aged, apparently healthy and body weight stable. Methods for measurements of blood lipids, serum insulin and in vivo insulin sensitivity have been described in detail[16].

**Clinical measurements**. All examinations took place at 8 am after an instructed overnight fast. Anthropometric and clinical chemistry variables were determined as described[15,16]. Total fat mass was calculated by bioelectrical impedance analysis. A WAT needle biopsy from the abdominal subcutaneous region lateral to the umbilicus was obtained each time. The samples were rinsed in sodium chloride (9 mg/mL) and portioned into two fractions; one 300 mg sample was frozen in liquid nitrogen and stored for gene expression analysis. The second fraction was used for adipocyte isolation to determine fat cell size as described[41].

**Cell culture**. Human adipose derived stem cells (hADSC) were isolated from the subcutaneous WAT as described and used previously[42]. hADSC were cultured and passaged in Dulbecco's modified Eagle's medium (DMEM) with 1 g/L of glucose (31885-023, Gibco, 10% FBS (SV301060.03, GE Healthcare), FGF₂ (2.5 ng/mL) (Sigma-Aldrich), penicillin/streptomycin (100 U/mL) and maintained at 37 °C in a humidified gassed incubator at 5% CO₂. For adipocyte differentiation, hADSC were cultured in DMEM/Ham's F12 media (21765-029, Gibco) supplemented with 10 μg/mL transferrin (Sigma-Aldrich), 0.86 μM insulin (Sigma-Aldrich), 0.2 nM triiodothyronine (Sigma-Aldrich), 1 μM dexamethasone (Sigma-Aldrich), 100 μM isobutyl-methylxanthine (Sigma-Aldrich), and 1 μM rosiglitazone (Caymen Chemical). After three days, dexamethasone and isobutyl-methylxanthine were removed; after nine days rosiglitazone was removed. Knockdown of the target lncRNA (ADIPINT) was carried out on day 8 of differentiation (when the cells showed adipocyte features), using a Neon™ transfection system MPK5000 (Invitrogen) with 2 pulses of 20 ms at 1300 V. Antisense LNA GapmeRs (Qiagen) and siRNAs (Dharmacon) used for ADIPINT and PC knockdown and control conditions are given in Supplementary Methods Table 1. Cell viability was assessed using Alamar blue fluorescence (ThermoFisher) as per manufacturers' recommendations. For PC inhibition experiments, Sodium Oxalate (Sigma-Aldrich) was added two hours post-transfection and Avidin (Sigma-Aldrich) on day 8 of differentiation. All

hADSC analyses were performed at day 13 of differentiation apart from lipid synthesis on day 14. For RNA quantification, lipolysis, triglyceride content, lipid synthesis and pyruvate measurements hADSC were cultured in 24-well plates at $2 \times 10^5$ cells/well. For bioenergetic quantification hADSC were plated on a Seahorse XF Microplate (Agilent) at $1.6 \times 10^4$ cells/well. For transcriptomics, quantitative proteomics and measurement of PC activity, hADSC were cultured in 6-well plates at $4 \times 10^5$ cells/well. For measurement of oxaloacetate hADSC were cultured in 10cm² cell culture dishes at $2.2 \times 10^6$ cells.

**Gene expression analysis**. Microarray analysis of coding genes in the WAT has been detailed[16]. Briefly, total RNA was extracted from frozen WAT with the RNeasy kit (Qiagen) and subjected to global transcriptome analysis using Clariom™ D arrays (Affymetrix) according to the manufacturer's instructions. The WT Plus Kit (Thermo Fisher) was used to amplify and biotinylate sense strand target cDNA before 5.5 μg was fragmented and hybridized to the arrays in a GeneChip Hybridization Oven 645 at 45 °C for 16–18 h. Arrays were washed and stained in a GeneChip Fluidics Station 450 prior to scanning in a Affymetrix GeneChip Scanner 7. Microarray gene expression analysis in hADSC after *ADIPINT* knockdown was carried out as described above; apart from that RNA was isolated using a NucleoSpin RNA Isolation Kit (Machery-Nagel). CEL files were preprocessed and analyzed using the Affymetrix Expression Console (version 1.4.1) and standard SST-RMA method. Affymetrix probe IDs were mapped to the FANTOM-CAT gene model[7], 54,980 probe sets mapped to 40,590 unique gene IDs. Qlucore Omics Explorer (http://www.qlucore.com) software was used to compare arrays as detailed previously[16]. Genes annotated lncRNA were selected for analysis and a paired comparison was made across time points with the effect of sample eliminated. qRT-PCR analysis was carried out as described[6]. Total RNA was reverse transcribed with iScript complementary DNA synthesis kits (BioRad). Assessments of RNA levels were performed using SYBR-green (BioRad) assays. Probes targeting ADIPINT, PC, GAPDH, 18 S, B2M, MALAT1, NEAT1 and MT-ND5 are listed in Supplementary Methods Table 2. The $\Delta\Delta C_T$ method was used for normalization for all experiments apart from RNA analysis in whole of fractionated WAT where the $\Delta C_T$ method was used. Either 18 S or B2M were used as housekeeping controls in all experiments.

**PacBio-sequencing**. RNA extracted from isolated human mature adipocytes were used to map the full-length transcript using Pacbio iso-seq method. The sample was prepared as described in "Procedure & Checklist – Iso-Seq™ Express Template Preparation for Sequel® and Sequel II Systems" (PN 101-763-800 Version 02 (October 2019) https://www.pacb.com/wp-content/uploads/Procedure-Checklist-Iso-Seq-Express-Template-Preparation-for-Sequel-and-Sequel-II-Systems.pdf. Briefly, using SMRTbell Express Template Prep Kit 2.0 (Pacific Biosciences) together with NEBNext® Single Cell/Low Input cDNA Synthesis & Amplification Module and Iso-Seq Express Oligo Kit (Pacific Biosciences). 300 ng of total RNA was used for cDNA Synthesis followed by cDNA Amplification. The amplified cDNA then went into SMRTbell library construction. Finally, the sample was sequenced on Sequel II System using Sequel® II Sequencing Plate 2.0 and On-Plate Loading Concentration of 80 pM. PacBio data were processed and evaluated with several tools in SMRT Analysis (v2.2.0 and v2.3.0) and in-house pipelines at the-National Genomics Infrastructure (NGI) /Uppsala Genome Center Sweden. The transcripts mapped to ADIPINT loci were extracted and presented in the current study. The complete sequences of the 5' full length trascripts of ADIPINT is available upon request.

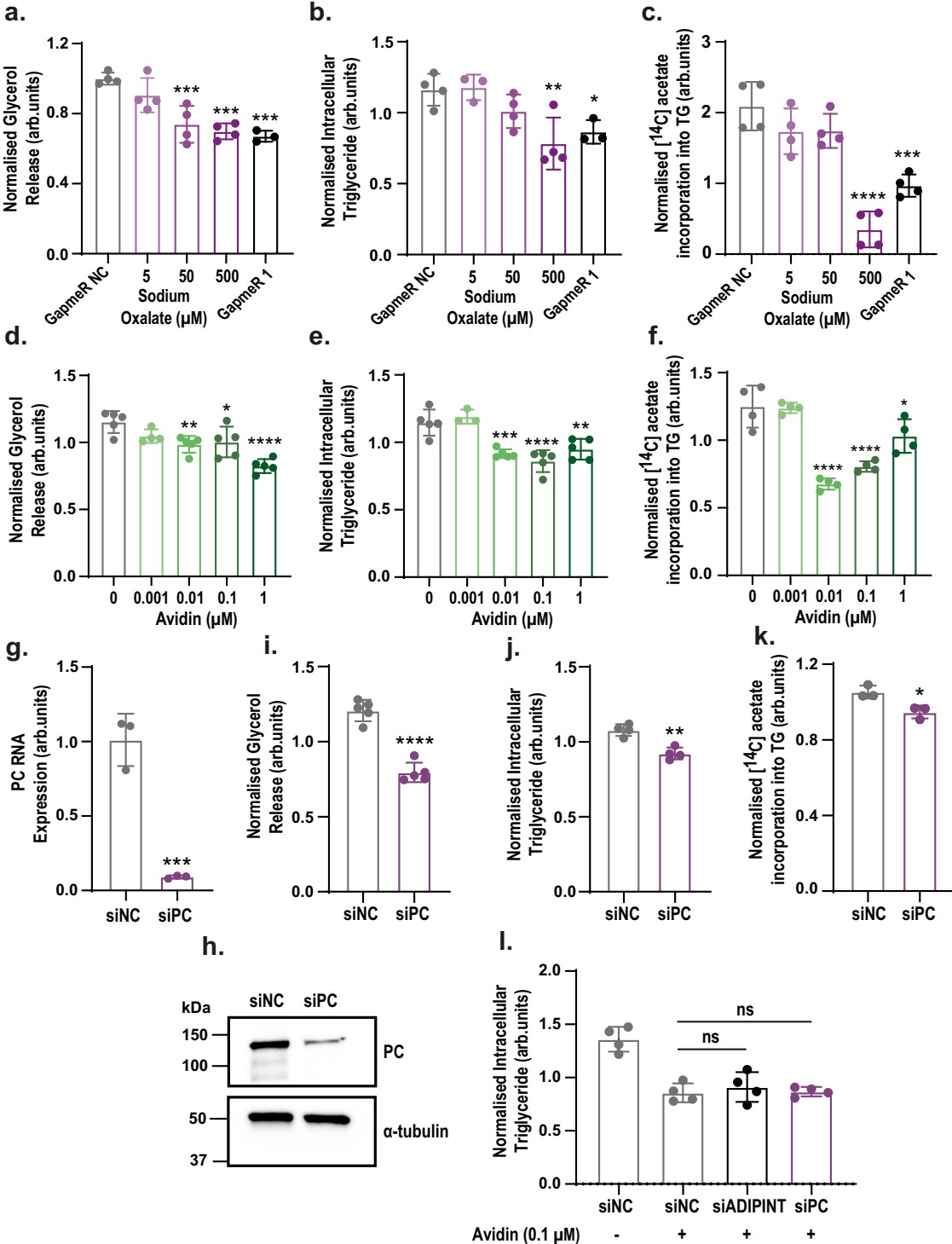

**Fig. 5 Pyruvate carboxylase inhibition recapitulates the effects on lipid metabolism as seen with ADIPINT knockdown. a** ($n = 3$-4), **d** ($n = 3$-5) Glycerol release (measure of basal lipolysis) **b** ($n = 3$-4), **e** ($n = 3$-5) intracellular triglyceride (TG) content and **c** ($n = 4$), **f** ($n = 4$) insulin-stimulated [$^{14}$C] acetate incorporation into intracellular lipid (measure of lipid synthesis) in differentiated hADSC incubated with varying concentrations of (**a**–**c**) oxalate or (**d**–**f**) avidin compared to vehicle treated cells. ADIPINT knockdown using GapmeR 1 was used as a positive control for the effects on glycerol release, triglyceride content and lipid synthesis in oxalate treated cells. **g** PC RNA ($n = 3$) and **h** protein expression following siPC knockdown compared with siNC transfected hADSC. **i** glycerol release ($n = 5$) **j** intracellular triglyceride content ($n = 4$) and **k** insulin-stimulated [$^{14}$C] acetate incorporation into intracellular lipid after PC knockown ($n = 3$). **l** The intracellular triglyceride content of hADSC after +\− avidin treatment and knockdown of either PC or ADIPINT. siADIPINT and siPC cells were compared with siNC cells treated with 0.1 µM avidin ($n = 4$). **a**–**g** and **i**–**l** each data point represents an individual experiment. One-way ANOVA with Dunnett's post-hoc test assessed significance. Data are presented as mean values +/− SD. *$p < 0.05$, **$p < 0.01$, ***$p < 0.001$, ****$p < 0.0001$. Source data are provided as a Source Data file.

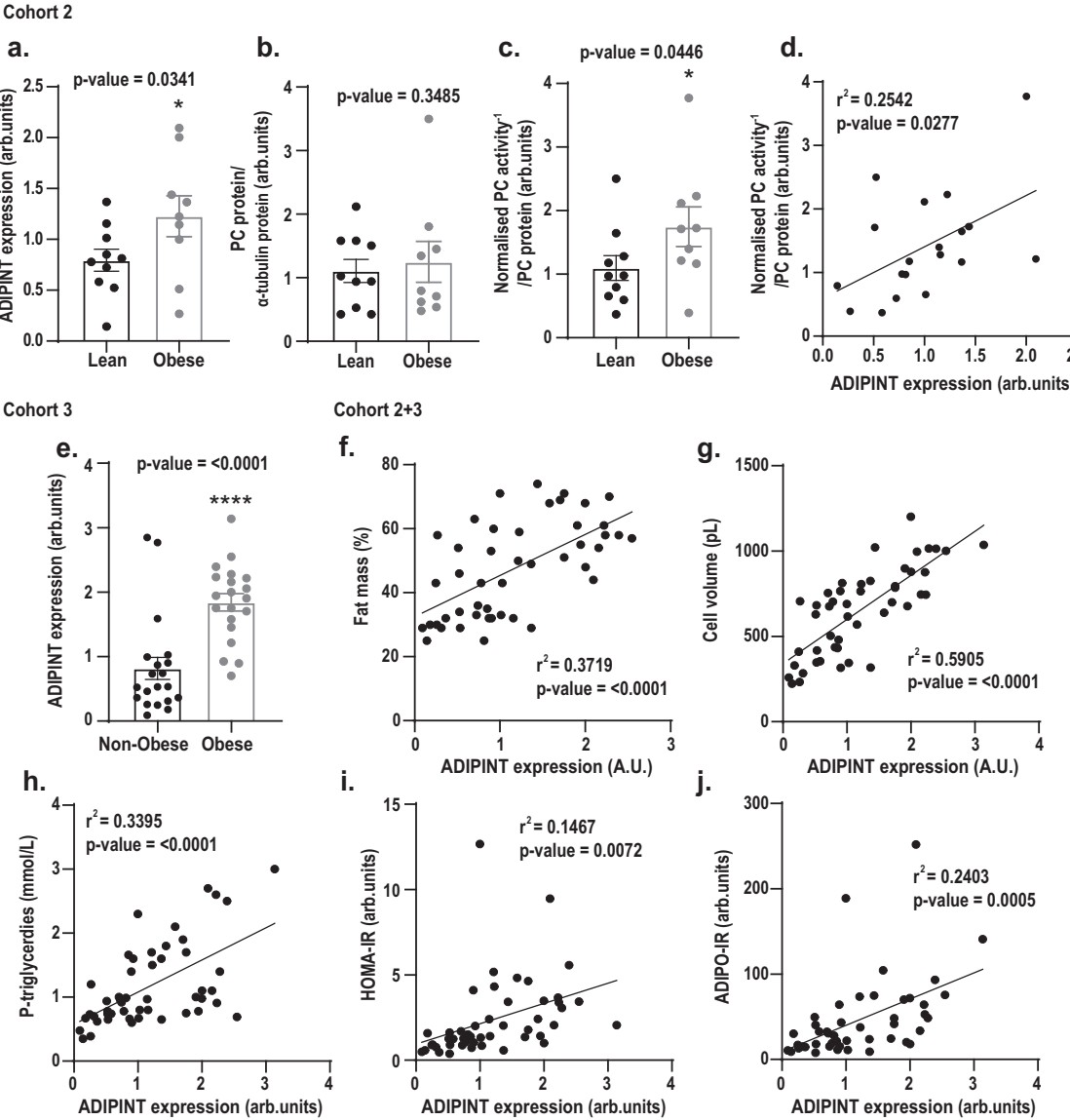

**Fig. 6 Subcutaneous adipose ADIPINT expression and PC activity correlate with each other and are increased in obesity.** Analyses of **a** ADIPINT expression, **b** PC protein and **c** PC activity in subcutaneous adipose tissue of 10 women without obesity (lean) and 9 women with obesity (Cohort 2). **d** ADIPINT expression plotted against PC activity in the same patient. A body mass index (BMI) of ≥30 kg/m² was used to distinguish individuals with obesity and BMI of <25 kg/m² for control. PC activity was normalized to PC protein expression from the same adipose tissue sample. **e** ADIPINT expression in subcutaneous adipose tissue of 20 women without obesity and 20 women with obesity (Cohort 3). A BMI of ≥ 30 kg/m² was used to distinguish individuals with obesity and BMI of <30 kg/m² for control. ADIPINT expression from both Cohort 2 and 3 plotted against **f** total percentage fat mass, **g** fat cell volume, **h** plasma triglycerdies **i** homeostasis model assessment of insulin resistance (HOMA-IR) and adipose tissue insulin resistance index (ADIPO-IR). Linear regression assessed significance in correlation analyses. A one-tailed t-test assessed significance between individuals with and without obesity. Each data point represents an individual patient, error bars represent SEM. Source data are provided as a Source Data file.

**Fluorescence microscopy**. RNA-FISH assays for ADIPINT using the View-RNA® ISH Cell Plus Assay Kit (Affymetrix) according to the manufacturer's instructions. Type 6 probes against ADIPINT were ordered from the same supplier. Briefly, hADSC were cultured in Millicell EZ SLIDE 4-well glass chambers (Merck Chemicals and Life Science AB). At day 13 of differentiation, the adipocytes were washed with phosphate buffered saline (PBS) and fixed in 4% paraformaldehyde for 20 min at room temperature. Subsequently, probes were hybridized for 2 h at 40 °C. After hybridization and amplification steps with pre-amplifier, amplifier, and linked labelled probe, the probes were detected using Alexa® Fluor 650 dyes, according to the manufacturer's instructions. Cells were stained with 1:2500 BODIPY 493/503 and Hoechst 33342 for 20 min before mounting with ProLong Gold Antifade Mountant (ThermoFisher). Images were obtained with a Nikon spinning disk confocal microscope system (Nikon) equipped with a Nikon ECLIPSE Ti inverted microscope and a 1.4 NA x60 oil immersion objective (Nikon), under the control of NIS-Elements AR version 4 software.

**Proteomic analysis**. Cell pellets were solubilized with 100 µL of 8 M urea and sonicated in a water bath for 10 min followed by further sonication using a VibraCell probe (Sonics & Materials, Inc.) for 40 s, with pulse 5/2, at 20% amplitude. Proteins were precipitated with 400 µL chilled acetone and incubated at −20 °C overnight. Protein concentration was measured by BCA assay (Thermo-Fisher) after centrifugation (16,000 x g) at 4 °C for 20 min and solubilizing proteins with 10 µL of 8 M urea and adding 10 µL ddH2O. 10 µg of protein in 253 µL of 133 mM ammonium bicarbonate (AmBic), pH 8 were sonicated in a water bath for 5 min and subjected to tryptic digestion following protein reduction in 5 mM dithiothreitol (Sigma) at 37 °C for 60 min and alkylation in 15 mM iodoacetamide for 60 min at room temperature in the dark. Trypsin (sequencing grade, Promega) was added in an enzyme to protein ratio of 1:20 (5 µL of 0.1 µg/µL) and digestion was carried out overnight at 37 °C. The digestion was stopped by adding formic acid at a final concentration of 5%. The samples then were cleaned on a C18 Hypersep plate with 40 µL bed volume (Thermo Scientific) and dried using a speedvac (MiVac, ThermoFisher).

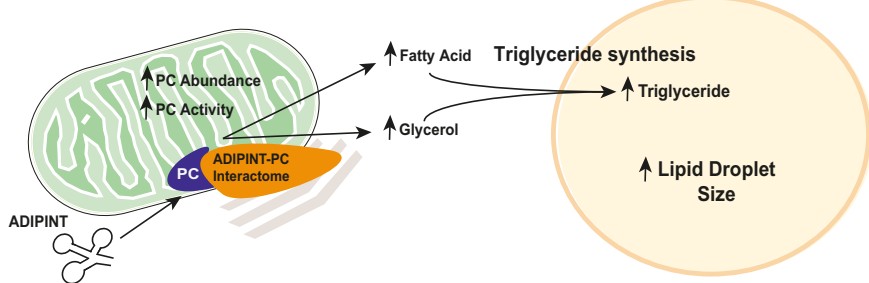

**Fig. 7 Current working model for the effect of the ADIPINT-PC interaction on fat cell metabolism.** ADIPINT interacts with PC regulating its interactome which results in an increased abundance and activity of PC within the mitochondria. The increase in PC activity leads to an increase in triglyceride synthesis in the adipocyte regulating the lipid droplet size.

Samples were dissolved in 70 μL of 50 mM triethylammonium-bicarbonate (TEAB), pH 8, and 40 μg of TMT-11 plex reagents (ThermoFisher) in 30 μL of dry acetonitrile were added. Samples were scrambled and then incubated at RT, 100 x $g$ for 2 h. The reaction was stopped by adding hydroxylamine at a final concentration of 0.5% and incubated at RT with 100 x $g$ for 15 min. Individual samples were combined into one analytical sample and dried in a speedvac, followed by cleaning on a C18 StageTip and re-dried in a speedvac.

Chromatographic separation of peptides was performed on a 50 cm long C18 reversed phase EASY-spray column connected to an Ultimate 3000 UPLC system (ThermoFisher). The gradient was run for 120 min at a flow rate of 300 nL/minutes and was as follows: 2–26% of buffer B (2% acetonitrile, 0.1% formic acid) for 110 min, up to 35% of buffer B for 10 min and up to 95% of buffer B for 2 min.

Mass spectra were acquired on an Orbitrap Fusion tribrid mass spectrometer (ThermoFisher) in 375 m/z to 1600 m/z at resolution of R = 120,000 (200 m/z) for full mass, targeting $4 \times 05$ ions, followed by data-dependent higher-energy collisional dissociation (HCD) fragmentations from precursor ions with a charge state 2+ to 7+ in 3 s cycle time. The tandem mass spectra were acquired with a resolution of R = 60,000, targeting $5 \times 104$ ions, setting quadrupole isolation width to 1.6 m/z and normalized collision energy to 30%.

The raw files were imported to Proteome Discoverer v2.3 (ThermoFisher) and analyzed using the SwissProt protein database with Mascot v 2.5.1 (MatrixScience Ltd) search engine. Parameters were chosen as follows: up to two missed cleavage sites for trypsin, peptide mass tolerance 10 ppm, 0.02 Da for the HCD fragment ions. Carbamidomethylation of cysteine was specified as a fixed modification, whereas oxidation of methionine and deamidation of asparagine was defined as variable modifications.

**hADSC lipid metabolism.** Basal (spontaneous) triglyceride hydrolysis (lipolysis) was quantified through medium glycerol concentration after three days incubation as described[43]. 50 μL of differentiation media was collected and added to 450 mL of resuspended ATP reagent (10 μL purified glycerokinase (Sigma-Aldrich) and 10 μL ATP Reagent SL (Biotherma) in 0.05 M Tris buffer). The decay in luminescence was measured over one minute and compared to a glycerol standard curve. Medium not exposed to cells was used as background. Stimulated lipolysis was quantified after a 3 h treatment of 1 μM isoprenaline (Sigma-Aldrich), the basal glycerol release over the same time period was subtracted from the isoprenaline stimulated glycerol release and presented. Cellular triglyceride content was determined using a triglyceride quantification kit MAK266 (Sigma-Aldrich). Each well was washed in PBS twice and cells were lysed in 5% NP-40 in ddH20. Fluorescence was measured at 590 nm on an Infinite M200 microplate reader (Tecan) and background was subtracted using the same volume of cell lysate minus the addition of the lipase enzyme as suggested by the manufacturer. A triglyceride standard curve was used to determine concentration. The lipid synthesis rate was measured through incorporation of [1-14C]-acetic acid (Perkin-Elmer) into acylglycerol lipids as described[44]. On day 13 of differentiation cells were cultured in DMEM media with no insulin, 10 μg/mL transferrin, 0.2 nM triiodothyronine and 1 μM glucose overnight. The following day, cells were cultured for 3 h in DMEM media containing 2 mM glucose, 10 mM HEPES, 0.2% BSA, 2 mM acetate and 1 mCi [1-14C]-acetic acid with or without 100 nM insulin. Cells were scraped in PBS with 0.1% SDS and radioactivity determined on a Tri-Carb 4910 TR scintillation counter (Perkin-Elmer). All lipid metabolic parameters were normalized to total protein content of the cells. To standardize each experiment, values were normalized to the average amount of glycerol/triglyceride/lipid per amount of protein for that differentiation.

**hADSC energy metabolism.** Bioenergetics were quantified using a XF96 Seahorse Extracellular Flux Analyzer (Agilent) as per manufacturers instructions. The Seahorse Glycolysis Stress Test Kit (Agilent) was used to assess OCR and ECAR at basal then after Glucose, Oligomycin and 2-deoxyglucose treatment. All concentrations used were as recommended by the manufacturer. Basal OCR and ECAR were carried out in medium supplemented with 2 mM glutamine. In the

individual experiments, OCR and ECAR for each well was first normalized to cell number by Hoechst 33342 staining of nuclei before counting using a CellInsight CX5 High Content Screening Platform (Fisher Scientific) and expressed as pmol/min/1000 cells for OCR and mpH/min/1000 cells for ECAR. In comparing independent experiments, each one was normalized by the average OCR or ECAR value for that run to minimize experimental variation.

**Targeted RNA identification using orthogonal organic phase separation (TROOPS).** Purification of the RNA-bound proteome was performed using orthogonal organic phase separation (OOPS) as described[20], and adapted by us for targeted pulldown of a specific RNA (TROOPS). hADSC plated at $3 \times 10^6$ cells in 15 cm$^2$ cell culture dishes were proliferated for 3 days (reaching ~90% confluence) before initiating differentiation. At day 13, cells were washed twice in ice-cold PBS and crosslinked at 254 nm (0.15 J/cm$^2$) on a UV Crosslinker 2400 (Stratagene). Cells were immediately scrapped in 1 mL Trizol (Invitrogen), transferred to a new 2 mL tube and left to stand for at least 5 min to allow unstable RNA-protein interactions to dissociate. 225 μL of chloroform was added to each tube, which was vortexed and spun for 15 min at 16,100 x $g$ at 4 °C. The upper aqueous phase (containing free RNA) and organic phase (containing free protein) was removed using a 25 G needle syringe. The interphase was washed in 1 mL Trizol followed by 200 μL chloroform twice, each time vortexed and spun before removal of aqueous and organic phases. Nine volumes of EtOH was added to the interphase and the sample stored at −20 °C for 30 min. Following incubation, the interphase was pelleted by centrifugation at 20,000 x $g$ for 20 min at 4 °C, the EtOH removed, the pellet gently washed in 80% EtOH and spun at 20,000 x $g$ for 15 min at 4 °C; this wash step was performed twice. The pellet was resuspended in 500 μL of 50 mM Tris-Cl (pH 7.0), 10 mM EDTA and 1% SDS with 1 mM PMSF, 0.1 U/μL SUPERase-In™ RNase Inhibitor (Invitrogen) and cOmplete Mini EDTA-free protease inhibitor cocktail (Roche). Two re-suspended pellets were pulled and sonicated using a Bioruptor® Pico (Diagenode) for 15 cycles (30 s on, 45 s off) with sonication beads. 1.75 mL of lysate (equivalent to $3.5 \times 15$ cm$^2$ culture dishes) was used per replicate in each TROOPS experiment. 15 μL of lysate was taken for protein and RNA input samples. Probe design and hybridization was modified from the ChirP-MS protocol[38]. For targeted pulldown, 24 biotinylated short oligos (Biosearch Technologies) were designed against *ADIPINT* and split into two pools termed odd and even. Each probe was numbered sequentially from 5' to 3', odd numbered probes were split into one pool and even numbers to another. A probe set designed against *Lac Z* mRNA was used as a control for non-specific probe interactions and odd and even probes added in the presence of RNase A (20 μg/mL) (Sigma-Aldrich) to control for the RNA independent pulldown of the target probes. All probe sequences are given in Supplementary Methods Table 3. 3.5 mL hybridization buffer (50 mM Tris-Cl (pH 7.0), 750 mM NaCl, 1% SDS, 1 mM EDTA, 15% formamide, 1 mM PMSF, 0.1 U/μL SUPERase-In™ RNase Inhibitor and cOmplete Mini EDTA-free protease inhibitor cocktail) was added to each lysate and 250 pmol targeting probes. Hybridization was carried out overnight at 37 °C with rotation. 500 μL of pre-washed MyOne™ Streptavidin C1 Dynabeads™ (Invitrogen) were added the following day for 2 h at 37 °C with rotation. Beads were extracted on a Dynamag-15 magnetic strip and washed 5 times in 300 mM NaCl, 30 mM sodium citrate, 0.5% SDS and 1 mM PMSF. After the final wash, 10% was taken for RNA purification and 90% for protein identification using mass-spectrometry. RNA bead samples were re-suspended in Proteinase K buffer (10 mM Tris-Cl (pH 7.0), 100 mM NaCl, 1 mM EDTA, 0.5% SDS) with 5% (v/v) Proteinase K (20 mg/mL) (Ambion) added and incubated at 50 °C for 45 min with rotation. Samples were boiled at 95 °C for 10 min and cooled on ice before the addition of 1 mL Trizol and RNA purification using the miRNeasy Mini Kit (Qiagen). qRT-PCR analysis of *ADIPINT* as well as two non-targeted abundant RNAs *GAPDH* and 18 *S* to assess pulldown efficiency. Protein bead samples were re-suspended in 22 uL of PBS with 1 uL RNase A (20 μg/mL), 1 uL RNase H (10 U/μL) and 1 uL cOmplete Mini EDTA-free protease inhibitor cocktail supplemented in and incubated for 45 min at 37 °C. 5 μL of Laemmli Sample Buffer was added to each sample before heating to 95 °C for 10 min. Samples were run onto an SDS-

PAGE gel, zinc stained to confirm protein loading and individual lanes were cut for MS-analysis or gel was transferred to a PVDF membrane for Western blot analysis.

For MS-analysis, gel pieces were washed with 0.2 M ammonium bicarbonate (AmBic) containing 50% acetonitrile. Proteins were reduced with 20 mM dithiothreitol (DTT) in 100 mM AmBic for 60 min at 37 °C, followed by alkylation with 55 mM iodoacetamide in 100 mM AmBic for 20 min at 25 °C and digested with 0.2 µg trypsin (sequencing grade) in 0.2 M AmBic overnight at 37 °C. The tryptic peptides were acidified with formic acid and extracted twice using 0.1% formic acid in 50% and 100% acetonitrile, respectively. The solutions were dried and the peptides were cleaned up on C-18 Stage Tips (ThermoFisher).

The reconstituted peptides in solvent A (0.1% formic acid in 2% acetonitrile) were separated on a 50 cm long EASY-spray column (ThermoFisher) connected to an Ultimate-3000 nano-LC system (Thermo Scientific) using a 60 min gradient from 4–26% of solvent B (98% AcN, 0.1% formic acid) in 55 min and up to 95% of solvent B in 5 min at a flow rate of 300 nL/minutes. Mass spectra were acquired on a Q Exactive HF Orbitrap mass spectrometer (ThermoFisher) in 350 m/z to 1600 m/z at resolution of $R = 120,000$ (at 200 m/z) for full mass, followed by data-dependent higher energy collisional dissociation (HCD) fragmentations from the 17 most intense precursor ions with a charge state 2+ to 7+. The tandem mass spectra were acquired with a resolution of $R = 30,000$, targeting $2 \times 10^5$ ions, setting quadrupole isolation width to 1.4 m/z and normalized collision energy to 28%.

Acquired raw data files were analyzed using the Mascot search engine v.2.5.1 (Matrix Science Ltd). Maximum of two missed cleavage sites were allowed for trypsin, while setting the precursor and the fragment ion mass tolerance to 10 ppm and 0.02, respectively. Carbamido-methylation of cysteine was specified as a fixed modification. Dynamic modifications of oxidation on methionine, deamidation of asparagine and glutamine and acetylation of N-termini were set. Initial search results were filtered with 5% FDR using Percolator to recalculate Mascot scores. Abundance values from each of the three TROOPS experiments were combined and analysed statistically to identify enriched protein(s).

**RNA immunoprecipitation (RIP).** hADSC were plated in 15 cm² cell culture dishes, differentiated until day 13 and UV-crosslinked as described for TROOPS using one 15 cm² cell culture dish per condition. Immediately there after cells were scraped in 1 mL RIP lysis buffer (50 mM Tris-Cl (pH 7.0), 150 mM NaCl, 10 mM EDTA, 1% Triton X-100, 1 mM DTT, 0.1 U/µL SUPERase-In™ RNase Inhibitor and cOmplete Mini EDTA-free protease inhibitor cocktail). Samples were passed through a 23 G syringe 20 times and spun at 12,000 x g for 10 min at 4 °C. The lysate was removed and cleared with 30 uL of Pierce™ Protein A/G magnetic beads (Thermo Scientific) for 30 min on rotation at room temperature. Magnetic beads were removed, and the sample spun once more at 12,000 x g for 10 min at 4 °C and lysate transferred to a fresh tube. Seven µg of rabbit pyruvate carboxylase (PA5-50101, ThermoFisher) or rabbit IgG (PP64B, Millipore) antibody was incubated with the lysate overnight at 4 °C with rotation. The following day 1:10 w/v Protein A/G beads were added to each sample and incubated for 3 h at 4 °C with rotation. Magnetic beads were extracted using a Dynamag-2 magnetic strip and washed for 5 min five times in RIP lysis buffer on rotation at room temperature. Beads samples were processed for RNA purification as described for TROOPS.

**Size exclusion chromatography.** ADIPINT or ADIPINT anti-sense cDNA was cloned out of a plasmid containing the full length cDNA sequence for ADIPINT transcript 2 (MICT00000363691.1) using KOD Hot Start Polymerase (Sigma-Aldrich) as previously described[45]. The PCR product was confirmed by size and EcoRI-HF (NEB) restriction digest, bands were excised from the agarose gel and purified using the NucleoSpin Gel and PCR Clean-up kit (Machery-Nagel). The RNAMaxx Transcription Kit (Agilent) was used as per manufacturers instructions to synthesize RNA, with 1 µg of template DNA product used per reaction. The reaction was carried out for two hours at 37 °C before RNA was purified using the NucleoSpin RNA Isolation Kit (Machery-Nagel). For RNA-PC binding, RNA was added to purified bovine PC (0.3 µg/µL Sigma-Aldrich) in a 110 uL reaction containing 10 mM HEPES (pH 7.3), 20 mM KCl, 1 mM MgCl₂, 1 mM DTT, 5% Glycerol and 0.1 U/µL SUPERase-In™ RNase Inhibitor and left at room temperature for 30 min. For experiments establishing the elution volumes for ADIPINT, PC and ADIPINT-PC, ADIPINT and PC were loaded at 1:1 molar ratio. When comparing ADIPINT to ADIPINT anti-sense, RNA was added to PC at a 0.03:1 molar ratio and DTT was omitted from the reaction. 100 µL of each reaction was then injected to a Superose® 6 Increase 10/300 GL column (Cytiva) using the Äkta go system (Cytiva) with the running buffer (pH 7.3), 20 mM KCl, 1 mM MgCl₂, 5% Glycerol and 0.01 U/µL SUPERase-In™ RNase. 250 µL of the indicated peak fractions (at 280 nm) identified in the chromatogram were collected and concentrated using Vivaspin® 500 (Sartorius) centrifugal concentrators (100 K MWCO, PES) for western blot analysis.

**Immuno-gold negative stain.** For sample preparation, ADIPINT or ADIPINT anti-sense were added to purified PC at a 1:1 molar ratio in 500 uL reaction and incubated at room temperature for 30 min before separating by the Superose 6 column (as described above). The void volume fractions were collected and concentrated using Vivaspin® 500 (Sartorius) centrifugal concentrators (50 K MWCO, PES). The protein concentration (around 0.03 mg/ml) was estimated via UV

absorption using a Nano Photometer (Implen, NP80, A260/280 is between 2–3). For ADIPINT anti-sense and PC control experiments, the void volume factions were directly concentrated to the same volume as in ADIPINT-PC. The PC antibody (16588-1-AP, Proteintech) was conjugated with colloidal gold particles (20 nm) according to the manufacturer's protocol (228-0005, NovusBio). Protein A-gold (10 nm) beads for the labeling of the anti-digoxin were purchased from electron microscopy sciences (EMS). In house 300-mesh carbon film coated copper grids were used for the immuno-negative stain experiments. The grids were all glow discharged with a PELCO easiGlow™ system prior to use (40 s, 20 mA). After sample application the grids were blocked with 0.1% BSA (Thermo Fisher) in wash buffer (10 mM HEPES, 100 mM KCl, pH 7.5). Each staining was repeated three times after the optimization for imaging.

For anti-PC immune labeling. 3.5 µL of ADIPINT-PC complex was applied onto the grids for 30 to 60 s. After the excess liquid was blotted away with a filter paper. The grids were blocked with 0.1% BSA (Thermo Fisher) for 30 min, before incubating with the conjugated anti-PC antibody for 1 h. After washing four times with washing buffer grids were immediately stained with 1 % uranyl acetate for 30 s, blotted and kept at room temperature for air drying before imaging.

For dual labeling, we used digoxin labeled ADIPINT (synthesized as described above with the addition of the digoxigenin-11-UTP (3359247910, Sigma-Aldrich) for the PC binding assay. After SEC and concentrating, the complexes were applied on grids and blocked with BSA for 30 min before as above. Then the digoxin antibody (25P1C9, Thermo Fisher) was applied for 1 h incubation. The grids were then incubated with gold-labeled protein A for 45 min Subsequently the grids were washed 3 times as above and then incubated with the gold labeled anti-PC antibody for 1 h incubation. Finally, the grids were washed again 3 times and stained by 1% uranyl acetate for imaging.

All electron micrographs were collected on a JEM 2100 f (JEOL) transmission electron microscope operated at 200 keV with a TemCam-XF416 CMOS camera (TVIPS). Images were collected under a nominal magnification of 60 K except for half RNAs (15 K) and with a defocus around 2–4 µM. Images were then processed by EM Measure (TVIPS) and analysed with ImageJ[46]. For quantitative analysis of gold particles on truncated RNAs, five representative images were used for each half RNA-PC sample and counted and processed by EM Measure.

**Cellular fractionation.** For fractionation of the nuclear and cytoplasmic compartments the RNA subcellular isolation kit (25501, Active Motif) was used as per manufactures instructions. Briefly hADSC cells cultured in 1 x 15cm² cell culture dish was differentiated to day 13 and washed in cold PBS before being scraped in complete lysis buffer provided by the kit. Cells were passed through a 23 G syringe 10 times and left for 20 min on ice to allow for sufficient lysis. The lysate was split in two, one aliquot for total RNA and the other for nuclear and cytoplasmic RNA isolation. The samples were washed and loaded onto the columns as instructed. *GAPDH* and *NEAT1* RNA were used to validate the cytoplasmic and nuclear fractions, respectively.

For fractionation into mitochondrial/membrane, cytoplasmic and lipid droplet compartments, hADSC cells cultured in 1 x 15 cm² cell culture dish were differentiated to day 13 and washed in cold PBS twice before being scraped in 750 µL homogenization buffer (20 mM Tris-HCl pH 7.4, 250 mM sucrose, 1 mM EDTA, 0.1 U/µL SUPERase-In™ RNase Inhibitor and cOmplete Mini EDTA-free protease inhibitor cocktail). Cells were passed through a 23 G syringe 10 times and spun at 1000 x g for 10 min at 4 °C removing nuclei and intact cells. The lysate was transferred to a fresh Eppendorf and spun at 20,000 x g for 40 min at 4 °C to pellet the mitochondrial/membrane fraction and lysate transferred to an ultracentrifugation tube (344057, Beckman) and mixed 1:1 with 50% OptiPrep Density Gradient Medium (D1556, Sigma-Aldrich) diluted in homogenization buffer. OptiPrep gradient solutions of 22, 16, 12, 8, 5, 2, and 0% were layered on top. Samples were spun overnight at 150,000 x g at 4 °C in a SW55 Ti swinging bucket rotor (Beckman-Coulter). Fractions were collected with a pipette and RNA/proteins were precipitated using the Wessel and Flügge method[47]. Each sample was split into two before precipitation, RNA samples were resuspended in 750 µL Trizol and subject to RNA isolation as described above. Samples for protein analysis were resuspended in RIPA Lysis and Extraction Buffer (89900, ThermoFisher) with a final concentration of 2% SDS. Western blot analysis was carried out as described above with the following antibodies used to stain for GAPDH (14C10, CST, 1:1000), GLS (ab200408, Abcam, 1:5000) and TOM20 (CST, 1:1000). All blots were incubated with the primary antibody overnight at 4 °C and the rabbit/mouse secondary antibody α-rabbit IgG, HRP-linked antibody (7074, CST 1:10000)/α-mouse IgG, HRP-linked Antibody (7076, CST 1:2500) for 1 h at room temperature.

**Pyruvate carboxylase (PC) activity assay.** PC activity was measured using the coupled enzyme reaction with malate dehydrogenase as described[23] with the following modifications. Cells were rinsed in cold PBS twice before being scraped in 180 µL assay buffer (10 mM HEPES (pH7.4), 250 mM sucrose, 2.5 mM EDTA, 2 mM cysteine and 0.02% BSA). Scraped cells were passed through a 23 G needle 10 times and sonicated using a Bioruptor® Pico for 15 cycles (30 s on, 45 s off) with no sonication beads added. Sonicated samples were spun at 11,000 x g for 10 min at 4 °C and 20 µL of lysate was saved for total PC quantification using western blot. Briefly, 5X Laemmli sample buffer was added to each sample, heated at 90 °C for 7 min, before loading onto a 4–12% SDS-PAGE gel. After running, the protein was transferred to a PVDF membrane using the Trans-Blot® Turbo™ Transfer System (Bio-

Rad) as per manufacturer's instructions. The membrane was blocked and incubated with rabbit pyruvate carboxylase (PA5-50101) or house-keeping α-tubulin (2144, CST) antibody overnight at 4 °C. The following day the membrane was washed with TBS-Tween 20 (0.1%) and incubated with α-rabbit IgG, HRP-linked antibody (7074, CST) for 1 h at room temperature. The membrane was washed in TBS-Tween 20, and the protein signal developed using the ECL™ Select Western Blotting Detection Reagent (GE Healthcare). Bands were visualized on a ChemiDoc MP Imaging System (Bio-Rad). Image J software was used to measure densitometry of PC and α-tubulin bands. 30 μL of the remaining lysate was added to 225 μL of PC activity buffer (80 mM Tris-Cl (pH 8.0), 2 mM ATP, 8 mM sodium pyruvate, 21 mM KHCO$_3$, 9 mM MgSO$_4$, 0.16 mM acetyl-CoA, 0.16 mM NADPH and 30 units of purified malate dehydrogenase (LMDH-RO Roche)). The absorbance at 340 nm (Absorbance of NADH) is read immediately, every 15 s for 4 h on a Varioskan™ LUX multimode microplate reader (Thermo Scientific). PC activity was calculated in two ways. Firstly, as the time taken to reach 50% completion of the reaction (½V$_{max}$) using the non-linear regression method with the Michaelis-Menten equation applied in GraphPad Prism 8.0. Secondly, we calculated it as initial velocity measured 20 min after starting the reaction when the changes in absorbance were in the linear range and expressed as nmol.min$^{-1}$.mL$^{-1}$. The relationship between PC concentration and ½V$_{max}$ was determined as linear (R$^2$ = 0.9204) by titrating down GapmeR NC transfected lysate and running both PC western blot and activity assay on the same sample (Supplementary Data Fig. 7a, b). Each ½V$_{max}$ and PC densitometry value was normalized to the average value for all the samples in that experiment to allow comparison between experiments ran on different days. A no-pyruvate control was ran showing no changes in the absorbance at 340 nm in the absence of pyruvate (Supplementary Data Fig 7c). For measurement in the mitochondrial fraction, cells were scraped in assay buffer and fractionated as stated above. The mitochondrial pellet was resuspended in assay buffer and activity measured. For measurement in WAT, 100 mg pieces were incubated on dry ice with 2 ×5 mm ball bearing before 1 mL of assay buffer was added and tissue lysed using a Tissue Lyser LT (Qiagen) at 50 oscillations/second for 10 min. 30 μL of lysate was saved for total PC quantification using western blot. The PC activity was then determined as stated above. Another 100 mg piece from the same biopsy was incubated on dry ice with 1 × 5 mm ball bearing before 1 mL of Trizol at oscillations/second for 5 min and subjected to RNA isolation as stated above.

**Oxaloacetate and pyruvate measurements**. Oxaloacetate and pyruvate concentrations were measured using the commercially available Sigma-Aldrich kits (Cat No. MAK070 and MAK071, respectively). Cells were washed in ice-cold PBS before being scraped in 200 μL of assay buffer and loaded onto a 10 kDa MWCO spin filter column (ThermoFisher) and spun at 13,000 x g for 15 min at 4 °C. Lysate passed through the column was then used for the oxaloacetate or pyruvate measurement as per the kits recommendations.

**PC Immunoprecipitation**. hADSCs grown in 1 × 15 cm$^2$ dishes per condition were washed twice in cold PBS and scraped in RIP lysis buffer without 1% Triton X-100 or 1 mM DTT. The mitochondrial/membrane fraction was isolated as stated above and resuspended in RIP lysis buffer without 1 mM DTT. The immunoprecipitation was carried out as stated for the RIP with the following modifications. 3 ug of PC antibody (ProteinTech) or rabbit IgG antibody was incubated with the lysate overnight at 4 °C with rotation. The following day 30 uL of Protein A/G beads were added to each sample and incubated for 30 min with rotation. Magnetic beads were extracted using a Dynamag-2 magnetic strip and washed for 5 min five times in Pierce™ IP lysis buffer (ThermoFisher) on rotation at room temperature. Bead samples were processed for mass-spectrometry as described for TROOPS.

**Statistical analyses and reproducibility**. GraphPad Prism v8.0 was used for all statistical comparisons unless otherwise stated. The specific statistical test used for each comparison mentioned in the figure legend. All data are expressed as mean ± S.D for experiments with sample size $n < 6$ and S.E.M when n was higher. All n represent biologically independent experiments. For ANOVA, Dunnett post-test was used for comparisons of all samples to a particular sample and Holm-Sidak was used for comparisons of all samples to one another. Comparisons between *ADIPINT* knockdown and control samples in microarray and proteomic studies were carried out using the Bioconductor R package Limma[48]. An FDR ≤ 0.05 was used to select for significant genes and FDR ≤ 0.1 for proteins in all comparisons. Pearson correlation was used to assess significance between the change in *ADIPINT* expression versus weight/adipocyte cell volume and *ADIPINT* versus PC mRNA expression at all time points. A *p*-value of ≤0.05 was considered significant. All representative western blot, electron micrograph and immunofluorescence images shown were performed at least three times.

**Reporting summary**. Further information on research design is available in the Nature Research Reporting Summary linked to this article.

## Data availability

Gene expression microarray data have been deposited at the NCBI Gene Expression Omnibus (GEO) and are identified by accession GSE199076 and GSE199063. Proteomics data are available via ProteomeXchange with identifier PXD032769. All data supporting the findings are provided in the Source Data File and other data are available from the corresponding authors on reasonable request.

## Code availability

All analyses were performed using the available toolboxes: R version 4.0.5 (http://www.r-project.org/), limma version 3.46.0 (https://bioconductor.org/packages/release/bioc/html/limma.html).

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

## Acknowledgements
The expert technical assistance of Katarina Hertel, Ana Maria Suzuki and Thais De Castro Barbosa is greatly acknowledged. This study was supported with grants from the Swedish Research Council (PA), Novo Nordisk Foundation (PA), the Strategic Diabetes Research Program at Karolinska Institutet (PA, HG), EMIF (Stockholm County) (PA) and the European Research Council (ERC) Synergy Grant SPHERES (agreement No. 856404) under the European Union's Horizon 2020 research and innovation programme (erc-spheres.univ-tlse3.fr) (DL). A Human Frontier Science Program grant ´(RGY0074/16) awarded to C.M. The China Scholarship Council (CSC) program awarded to Z.W. AG Kerr, KHM Kwok and A Ludzki were sponsored by the Novo Nordisk Postdoctoral Fellowship program at Karolinska Institutet. Protein identification, quantification, in-gel digestion, mass spectrometric analysis and database searches for protein identification was carried out at the Proteomics Biomedicum core facility, Karolinska Institute, Stockholm. (https://ki.se/en/mbb/proteomics-biomedicum). RNA sequencing and microarray was carried out at BEA core facility at Karolinska Institute, Stockholm (https.bea.ki.se). We thank the Live Cell Imaging facility, Karolinska Institutet, Stockholm (https://ki.se/en/bionut/welcome-to-the-lci-facility) for microscopy imaging analysis. We would like to acknowledge support of the National Genomics Infrastructure (NGI) /Uppsala Genome Center and UPPMAX for providing assistance in massive parallel sequencing and computational infrastructure. Work performed at NGI / Uppsala Genome Center has been funded by RFI / VR and Science for Life Laboratory, Sweden.

## Author contributions
A.G.K., H.G. and P.A. planned the study, designed experiments and analyzed the data. A.K.G., H.G., N.W., J.J., K.H.M.K, and A.L. performed experiments. A.G.K. and H.G. developed TROOPS and D.L. developed the lipid synthesis assay. Z.W. and C.M. planned, performed, and analysed the SEC and electron microscopy experiments. M.B. advised and planned Seahorse experiments. P.A. recruited and phenotyped the clinical cohorts. I.D. provided microarray data on WAT. P.A., A.K.G., and H.G wrote the first version of the paper. All authors contributed to further writing and approved the final version.

## Funding
.

## Competing interests
The authors declare no competing interests
