## [Peer Review File · Nature Communications]

REVIEWER COMMENTS

Reviewer #1 (Remarks to the Author):

The manuscript "The long noncoding RNA ADIPINT is a gatekeeper of pyruvate carboxylase function regulating human fat cell metabolism" examines an lncRNA ADIPINT as a regulator of lipid metabolism, by binding and regulating pyruvate carboxylase enzyme in mitochondria. ADIPINT knockdown via RNA interference lowers metabolic activity according to several measures. A novel pulldown strategy to examine ADIPINT protein binding partners identified pyruvate carboxylase. ADIPINT also appears to encourage mitochondrial protein binding of pyruvate carboxylase to other proteins involved in glycolysis. Overall, the manuscript is very well written, and they carried out right experiments to support their hypothesis. While I am supportive of this manuscript, a few additional clarifications listed below would improve the manuscript quality even further.

Major comments:

1. I am unclear on the selection of candidates, female only candidates were used? Is it expected that gastric bypass alters lncRNA expression through an indirect means based on fat reduction, or does lncRNA expression regulate fat storage and metabolism (in which case what is the function of gastric bypass)? Why was the 5-year data excluded from analysis? A sentence to explain this would be helpful to readers less familiar with the specifics of the model.
2. For the comprehensive picture on the molecular association and integration of ADIPINT and PC metabolome, some structural experimental evidence would be needed.

Minor comments:

1. Page 4, line 88: ADIPINT instead of APIPINT.
2. Page 5, line 132: GAPDH instead of GADPH
3. Page 6, line 143-144 "RNA-FISH with immunofluorescence staining for ADIPINT and PC in tandem confirmed the localization overlap (Fig. 3h). Together these data suggest a physical ADIPINT-PC interaction". I cannot see this overlap in the immunofluorescence images.
4. ADIPINT is represented as little yellow dots, while PC has a red signal distributed everywhere. Authors should verify the ADIPINT signal in knockdown cells to verify the specificity of the signal and maybe add another mitochondrial marker, like TOMM20, to verify the localization of ADIPINT at the mitochondrial level.
5. Page 7, line 170: immunoprecipitated instead of immunoprecipiated.
6. Authors should specify if figure 6b is a protein level, as well as the figure should be supported by the representative Western blot data.

Reviewer #2 (Remarks to the Author):

At the core of this manuscript, Kerr et al have a remarkable finding. They identify a protein-lncRNA interaction with a novel methodology, and it appears that this interaction has metabolic consequences. However, in spite of this potentially exciting + impactful data, much of the physiology and enzymology presented are flawed. New experiments and new analyses are needed.

1. Big picture: the results are very interesting, but does the interpretation make sense? Essentially, the authors are saying that with weight loss, ADIPINT decreases, thereby decreasing the functionality of PC in the adipose tissue. Are the authors suggesting that this lncRNA evolved to be most functional in the setting of nutrient oversupply? A careful rewrite of the Results and Discussion would help the reader to understand better the authors' take on the big picture.

2. Figure 2 is important for this manuscript but has a number of problems.

a) Neither basal lipolysis nor cellular triglyceride content appear to be normalized to something relevant (viable cell number, cell mass, or protein content).

b) Differences in "lipolysis index" alone are difficult to interpret. It would be useful to see if the lipolysis index responds differently to basal media vs high insulin or basal media vs beta 3 agonist added.

c) Normalized OCR and normalized ECAR should have real units (e.g. pmol/(min*mg protein)), not arbitrary units.

d) Did the transcriptomic and proteomic analyses find any adipocyte-identifying genes/proteins that increased after treatment? Does the ASO treatment uniformly decrease adipocyte-identifying genes?

e) If ADIPINT ASO is in any way toxic to the adipocytes (if ADIPINT is as important as the authors propose, the ASO may very well be toxic), you could see decreased lipolysis, decreased triglyceride, decreased OCR, decreased ECAR, (and possibly decreased adipocyte identity). The authors need to demonstrate that this treatment is not simply toxic.

3. The kinetic experiments need to be reanalyzed.

The use of "50% completion" would be a reasonable surrogate marker for enzyme activity if the changes in velocity were large, and the activity experiments were not important to the point of the manuscript. In this case, the changes in velocity are small, and the activity experiments are quite important. The "completion" point as determined by absorbance at 340 nm depends on factors outside the investigator's control, and the completion point is not entirely due to PC activity. Please report activity calculated from initial velocities.

Additionally, it is not clear that a "no pyruvate" control was used. Every run on the spectrophotometer requires a "no pyruvate" control, so the authors can subtract out differences in disappearance of NADH unrelated to the enzyme.

Finally, if the authors could possibly use their synthetic ADIPINT to rescue the reduced PC activity, that would make this experiment much more convincing.

4. Line 158: "Thus, lower oxaloacetate production..." as far as I can tell, the authors didn't measure rates of OAA production. OAA concentration will change based on multiple fluxes, and these will not all be dependent on pyruvate concentration.

5. Oxalate is an excellent chelator, attaching to and so can be quite useful for studies of PC enzymology. I am skeptical about its use in figure 5. At the higher concentrations, oxalate will chelate magnesium and calcium (and iron), which will affect the cellular physiology of the adipocytes. Figure 5 should be repeated with a more specific PC inhibitor. If that isn't possible, it should be repeated with another chelator, to at least ensure that oxalate's magnesium-binding properties aren't the key to this

experiment.

6. Lines 261-262: “generating ATP necessary for lipolysis.” This isn’t true in any meaningful way. Certainly within the adipocyte, lipolysis is a hydrolysis reaction. The fact that this is wrong causes problems for the rest of the authors’ argument in this paragraph.

Minor points:

7. That ADIPINT appears to bind PC protein is central to this study. However, this is suddenly introduced on lines 129-130. Out of the handful of potential interactions identified, how did the authors pick this protein? Did they screen all of the hits from supplementary table 11 with pulldowns, and this was the only one that clearly bound ADIPINT (if so, please show this in the supplement)? Or did they pick it simply based on what is known already about PC function?

8. Line 149- the reference to figure 2e is clearly not right.

9. Take care with interpreting the activity data- while it is great that the authors see differences, suggesting that the ADIPINT-PC interaction is real, such small differences in specific activity are unlikely to drive a meaningful change in substrate flux in vivo.

10. How were intracellular oxaloacetate and pyruvate concentrations determined (fig 4d-e)? This needs to be described in detail- the assays for these intracellular metabolites are difficult and easily confounded.

11. The authors should probably decrease their discussion of the “cytosolic” PC fraction, as this seems like a distraction- PC activity is clearly localized to the mitochondria, and the authors don’t need to really understand what the cytosolic band is in order to tell us about the changes in the mitochondrial PC.

12. Supplementary table 13 should be divided into the two groups that were compared (we should be able to compare the obese and non-obese subjects).

13. Figure 6 needs to be interpreted with caution.

The differences in 6c, d, f, and h look as if they would disappear without a single (the highest) point. HOMA-IR correlates with hepatic insulin resistance- which correlates with fatty liver, which increases with obesity. If the authors want to say that there is an effect on adipose insulin resistance, they should try another measure (like Adipo-IR, explained well in PMID: 28465607).

14. Line 209: “These data support the notion that ADIPINT is a physiological regulator of lipid storage and fat cell size”. This statement is extremely speculative- if the authors really want to look at the relationship between ADIPINT and physiological lipid storage, that would be a whole new study in itself (perhaps looking at ADIPINT expression in equally obese individuals and comparing this with lipid inappropriately stored in non-adipose tissues).

15. Throughout the manuscript the authors talk about a “rate of lipid metabolism”, as if “lipid metabolism” was a single entity. This prevents a clear explanation of what is actually happening. Lipolysis, esterification / re-esterification, lipogenesis, and oxidation are all separate processes.

16. The English language could use some work. The authors haven’t paid attention to some of the basics

of English grammar, including very simple concepts like number and tense. While tense matching within a paragraph is often - usually - ignored in scientific writing, it still deserves attention within a sentence.

Reviewer #3 (Remarks to the Author):

Kerr and authors present a nice human and in vitro adipocyte cell biology study of a human adipocyte specific long non coding RNA, ADIPINT. They show that ADIPINT regulates pyruvate carboxylase (PC) in the mitochondria of human white adipocytes, an enzyme that plays a role in glycolysis and lipogenesis. Knockdown (KD) of ADIPINT reduced pyruvate carboxylase activity and its interactome (but not its protein level) and this corresponded with reductions in intracellular triglycerides and glycerol release from adipocytes, indicating perturbations in lipid metabolism. They also show that KD of ADIPINT reduced oxygen consumption rate and extracellular acidification rate using Seahorse indicating reduction in glycolysis. The authors present focused human translational data on ADIPINT. They show that ADIPINT is increased in the subcutaneous adipose tissue of obese women compared to lean women and corresponding increase in PC activity. ADIPINT expression correlated positively with adipocyte cell volume, plasma triglycerides, insulin and HOMA-IR which provides correlative support for ADIPINT in regulation of adipocyte lipid homeostasis and lipogenesis. An strength of the paper was identification of an interaction of PC with ADIPINT through a Targeted RNA-protein orthogonal organic phase separation (TROOPS), targeting a specific lncRNA within the after isolating the entire RNA-protein bound proteome. The TROOPS method seems like a valuable improvement on current RNA pull down methods. Overall, the authors present a potentially important finding relating to novel regulatory pathways in adipocytes and energy metabolism, especially in human – yet there are major limitations in its current form that need to be addressed

Overall, this study presents potentially important findings of this lncRNAs ADIPINT role in energy metabolism and also highlights that lncRNAs can have interactions with enzymes such pyruvate carboxylase, thus further expanding our understanding of lncRNA functions, particularly in adipocytes and energy metabolism. However, there are major deficiencies that should be addressed.

Major concerns:

1) There are several technical deficiencies in the lncRNA studies. The authors state that ADIPINT transcripts do not have protein coding potential according to CPAT, RNAcode, phyloCSF and sORF ribose coding prediction softwares. This should be validated experimentally e.g., by in vitro translation assay. Although CAGE-seq data is very useful, 5' or 3' RACE experiments should be used to confirm the full length and diversity of isoforms in human adipocytes. There is only one ADIPINT knockdown strategy (ASOs) and no complementary gain of function genetic studies. A second knockdown method should be used to confirm some of the findings? And gain of function genetic studies should be used to complement the knock-down loss of function phenotypes.

2) The pyruvate carboxylase (PC) inhibitor studies support the authors hypothesis but are far from conclusive. Only pharmacological approaches were used and no genetic loss-of-function or gain-of-function strategies were applied. Further, there were no significant effects at 5um or 50um, only at 500um, concentrations of oxalate – how specific is this high-concentration? These data need to be validated with genetic strategies. Furthermore, genetics and/or pharmacological data are needed to prove that that ADIPINT acts via PC in mediating effects on adipocyte lipid metabolism - specifically, to prove the mechanistic dependency e.g., through joint manipulations or rescue approaches. Thus, after knock down of PC, was there any effect of ADIPINT KD beyond that of PC? Could overexpression of ADIPINT rescues partial knockdown of PC?

3) Furthermore, there is little evidence in this paper or in published work for a causal role of PC in

adipocyte/adipose lipid metabolism and pathophysiology in vivo. In the absence of straightforward traditional in vivo mouse modeling of ADIPINT (because of lack of conservation), it would be appropriate to perform adipose-targeted genetic studies of PC in mouse in order to validate the relevance and potential role of PC in adipose pathophysiology in vivo. This is particularly important given the lack of work on the PC pathway in mouse models of obesity, fat distribution, depot storage, and insulin sensitivity etc – i.e., regarding key conclusions about PC and its regulation in human adipocytes by ADIPINT drawn by the authors based on adipocyte studies and human correlative human data.

4) What is the authors justification for only looking at female obese patients for this study? The lack of males in the discovery and validation human studies needs to be addressed. Further, although the design of human discovery studies is elegant and rigorous, the follow up studies in human are limited and somewhat confusing. Were the samples/data used in the follow up studies the same participants used in the discovery? If so, these are not independent and additional validation and clinical extension should be performed especially as noted above, studies that also include males, and ideally larger numbers than nine individuals.

5) Given the proposed role of ADIPINT in mitochondria, it is appropriate to perform some studies of ADIPINT in brown/beige/brite adipocytes – especially as noted in the context of lack of causal mechanistic studies in vivo.

6) The conclusions in discussion should be toned down. This work is a far distance from providing evidence that “ADIPINT-PC interactions could have an important clinical role and constitute a pharmacological target for metabolic diseases.”

Minor

1) Line 77: “Transcript 2 is 4.8 kb long and contains two exons (Fig. 1c); the junction was validated by qPCR and Sanger sequencing (data not shown).” The authors should show this qPCR data and a BLAT map of the amplicon that was sent for Sanger sequencing in supplement?

2) Supplement table 2 should contain the genomic location of the lincRNAs?

3) The FISH experiment Figure 1h. Can the authors include a merged image of HOECHST and ADIPINT only, as the BODIPY make it difficult to see the sub cellular location?

4) Did the Gapmer Knockdown effect differentiation of adipocytes (Figure 2)?

5) Line 149: ‘PC expression was not altered at the protein level after ADIPINT knockdown (Fig 2e)’ – There is no protein expression of PC in Fig2e. Please correct this and include in figures.

6) The n numbers for each experiment should be indicated in each figure legend.

7) Line 160: Typo “thetotal” needs a space ‘the total’

8) Line 207: Typo “fat cell volume and ADIPINT expression (Fig. 6i)” – Should be Fig.6e as displayed

REVIEWER COMMENTS

Reviewer #1 (Remarks to the Author):

The manuscript "The long noncoding RNA ADIPINT is a gatekeeper of pyruvate carboxylase function regulating human fat cell metabolism" examines an lncRNA ADIPINT as a regulator of lipid metabolism, by binding and regulating pyruvate carboxylase enzyme in mitochondria. ADIPINT knockdown via RNA interference lowers metabolic activity according to several measures. A novel pulldown strategy to examine ADIPINT protein binding partners identified pyruvate carboxylase. ADIPINT also appears to encourage mitochondrial protein binding of pyruvate carboxylase to other proteins involved in glycolysis. Overall, the manuscript is very well written, and they carried out right experiments to support their hypothesis. While I am supportive of this manuscript, a few additional clarifications listed below would improve the manuscript quality even further.

Response: We thank the reviewer for the constructive criticisms.

Major comments:

1. I am unclear on the selection of candidates, female only candidates were used? Is it expected that gastric bypass alters lncRNA expression through an indirect means based on fat reduction, or does lncRNA expression regulate fat storage and metabolism (in which case what is the function of gastric bypass)? Why was the 5-year data excluded from analysis? A sentence to explain this would be helpful to readers less familiar with the specifics of the model.

Response: In our clinical practice, women are easier to recruit than men for fat biopsies. This makes it difficult to make a valid matching of men/women in prospective studies. We have now added a third group of females showing the same results as with the original cohort (Figure 6 and Extended Data Figure 6). Thus, we feel confident about our finding of ADIPINT in women and have added in the discussion that results may be different in men. We have also added the five-year data to the manuscript (Figure 1a-b).

The function of gastric bypass is as a model to examine the expression of lncRNA before and after weight loss and in relation to changes in the white adipose tissue phenotype. What exactly drives the lncRNA expression changes we have not examined in this study, though based on our data, it is possible that changes in ADIPINT expression following bariatric surgery may contribute to the altered metabolism of the adipocyte. This is better dealt with in the revised MS and further investigation are needed to infer causality.

2. For the comprehensive picture on the molecular association and integration of ADIPINT and PC metabolome, some structural experimental evidence would be needed.

Response: We have now added negative staining electron micrographs visualizing the detailed molecular binding of ADIPINT to PC (Figure 4g, h, Extended Data Figure 4c-j). To identify the region of ADIPINT needed for the PC interaction, we have added experiments with truncated versions of ADIPINT (Figure 4i-l). This includes size exclusion chromatography and further electron micrographs using the truncated ADIPINT.

Minor comments:

1. Page 4, line 88: ADIPINT instead of APIPINT.

Response: Now corrected.

2. Page 5, line 132: GAPDH instead of GADPH

Response: Now corrected.

3. Page 6, line 143-144 "RNA-FISH with immunofluorescence staining for ADIPINT and PC in tandem confirmed the localization overlap (Fig. 3h). Together these data suggest a physical ADIPINT-PC interaction". I cannot see this overlap in the immunofluorescence images.

Response: To better demonstrate the overlap in fluorescence signal for ADIPINT and PC, we have revised the presentation of the RNA-FISH images. We also added profile plots of pixel intensity for ADIPINT and PC at selected regions. Data are now in Figure 3m.

4. ADIPINT is represented as little yellow dots, while PC has a red signal distributed everywhere. Authors should verify the ADIPINT signal in knockdown cells to verify the specificity of the signal and maybe add another mitochondrial marker, like TOMM20, to verify the localization of ADIPINT at the mitochondrial level.

Response: We have now verified the ADIPINT signal using siRNA knockdown and added this to Extended Data Figure 1. In addition, we have also performed the co-localization analysis as done with PC for TOM20. The data was added to the Extended Data Figure 3n.

5. Page 7, line 170: immunoprecipitated instead of immunoprecipiated.

Response: Now corrected

6. Authors should specify if figure 6b is a protein level, as well as the figure should be supported by the representative Western blot data.

Response: Figure 6b is PC protein data, we have now clarified this in the figure legend and added a representative blot to Extended Data Figure 6b.

Reviewer #2 (Remarks to the Author):

At the core of this manuscript, Kerr et al have a remarkable finding. They identify a protein-lncRNA interaction with a novel methodology, and it appears that this interaction has metabolic consequences. However, in spite of this potentially exciting + impactful data, much of the physiology and enzymology presented are flawed. New experiments and new analyses are needed.

Response: We are very grateful for the constructive criticism.

1. Big picture: the results are very interesting, but does the interpretation make sense? Essentially, the authors are saying that with weight loss, ADIPINT decreases, thereby decreasing the functionality of PC in the adipose tissue. Are the authors suggesting that this lncRNA evolved to be most functional in the setting of nutrient oversupply? A careful rewrite of the Results and Discussion would help the reader to understand better the authors' take on the big picture.

Response: We have now rewritten the discussion with a clearer focus on the big picture. We believe the major function of ADIPINT is to build up triglycerides in fat cells. Independent of weight status (lean or obese) we find ADIPINT to be highly correlated with fat cell size when the subjects are in a caloric steady state (Figure 6f). Before, two and five years post-bariatric surgery when there are marked variations in body weight, we find the changes in fat cell volume to be highly correlated with changes in ADIPINT expression (Figure 1a, b). The dominant effect when we knock out ADIPINT is a reduction in intracellular triglyceride content of fat cells. Thus, we do not believe ADIPINT to be only functional in the setting of nutrient oversupply but to be functional whenever there are changes in the fat cell triglyceride content.

2. Figure 2 is important for this manuscript but has a number of problems.

a) Neither basal lipolysis nor cellular triglyceride content appear to be normalized to something relevant (viable cell number, cell mass, or protein content).

Response: All measures of basal lipolysis and triglyceride content are normalized to protein content for each experiment. To account for the variance per experiment in the total amount of glycerol release or triglyceride, each experiment is normalized to the average amount of glycerol or triglyceride measured that time. We now explain this more clearly in the methods section.

b) Differences in "lipolysis index" alone are difficult to interpret. It would be useful to see if the lipolysis index responds differently to basal media vs high insulin or basal media vs beta 3 agonist added.

Response: We thank the reviewer for the helpful suggestion, we have now added isoprenaline (non-selective beta agonist) stimulated lipolysis data to the manuscript (Extended Data Figure 1). Interestingly, ADIPINT knockdown only appears to affect the basal lipolysis as the stimulated lipolysis appears to function as control cells. We are not

surprised that ADIPINT is acting merely on basal lipolysis. This is also true for other important regulators of human fat cell lipolysis such as adenosine and tumor necrosis factor alpha.

c) Normalized OCR and normalized ECAR should have real units (e.g. pmol/(min*mg protein)), not arbitrary units.

Response: OCR and ECAR values were normalized to cell number and all four experiments are plotted with pmol/min/1000 cells or mpH/min/1000 cells and shown in Extended Data Figure 2g, h. As described for glycerol and triglyceride values above, we normalized the values within each experiment to allow us to examine if there was a consistent difference between ADIPINT knockdown and control cells. We have now added an explanation of this to the methods section.

d) Did the transcriptomic and proteomic analyses find any adipocyte-identifying genes/proteins that increased after treatment? Does the ASO treatment uniformly decrease adipocyte-identifying genes?

Response: The transcriptomic and proteomic analyses did not find any adipocyte identifying (marker) genes/proteins increased after ADIPINT knockdown. We did not, however, see a uniform decrease in all adipocyte marker genes including ADIPOQ, PLIN1 and FABP4, which were unchanged at both the RNA and protein level after knockdown. This is clarified in the results text.

e) If ADIPINT ASO is in any way toxic to the adipocytes (if ADIPINT is as important as the authors propose, the ASO may very well be toxic), you could see decreased lipolysis, decreased triglyceride, decreased OCR, decreased ECAR, (and possibly decreased adipocyte identity). The authors need to demonstrate that this treatment is not simply toxic.

Response: We have now added Alamar blue fluorescence measurements of the cells after knockdown using the ADIPINT ASOs. We do not see any differences in the viability of the cells after ADIPINT knockdown. We have added this data to Extended Data Figure 2f.

3. The kinetic experiments need to be reanalyzed.

The use of “50% completion” would be a reasonable surrogate marker for enzyme activity if the changes in velocity were large, and the activity experiments were not important to the point of the manuscript. In this case, the changes in velocity are small, and the activity experiments are quite important. The “completion” point as determined by absorbance at 340 nm depends on factors outside the investigator’s control, and the completion point is not entirely due to PC activity. Please report activity calculated from initial velocities.

Additionally, it is not clear that a “no pyruvate” control was used. Every run on the spectrophotometer requires a “no pyruvate” control, so the authors can subtract out differences in disappearance of NADH unrelated to the enzyme.

Finally, if the authors could possibly use their synthetic ADIPINT to rescue the reduced PC activity, that would make this experiment much more convincing.

Response: We thank the reviewer for the advice on plotting our PC activity experiments. We have now plotted PC activity as initial velocity and added this to Figure 4a and Extended Data Figure 4c. We ran no pyruvate controls in the first experiments configuring the assay and, in our hands, found very negligible changes in NADPH over the time measured. We now include an example of the no pyruvate control compared with pyruvate added in Extended Data Figure 7c.

Regarding using synthetic ADIPINT to rescue PC activity, we do not believe that the major action of ADIPINT is to directly modulate the activity of PC as in the classical allosteric modulator. Our evidence that knockdown of ADIPINT changes the localization and interactome of PC leads us to believe this is the regulatory effect of ADIPINT on PC. Changes in the interactome of metabolic enzymes have been shown to alter their activity as shown here (PMID: 29849027). It may be the changes in velocity we observe are secondary to the above-described changes. We have now added a section on the discussion clarifying our interpretation of the results and emphasize that changes in PC activity per se is less important than the effect on PC localization/interactome after ADIPINT knockdown.

4. Line 158: “Thus, lower oxaloacetate production...” as far as I can tell, the authors didn’t measure rates of OAA production. OAA concentration will change based on multiple fluxes, and these will not all be dependent on pyruvate concentration.

Response: We have now corrected this to read lower oxaloacetate concentration.

5. Oxalate is an excellent chelator, attaching to and so can be quite useful for studies of PC enzymology. I am skeptical about its use in figure 5. At the higher concentrations, oxalate will chelate magnesium and calcium (and iron), which will affect the cellular physiology of the adipocytes. Figure 5 should be repeated with a more specific PC inhibitor. If that isn't possible, it should be repeated with another chelator, to at least ensure that oxalate's magnesium-binding properties aren't the key to this experiment.

Response: We have carried out the PC inhibition studies with a second inhibitor, Avidin and added this to Figure 5d-f. We have also carried out PC knockdown studies using siRNA to confirm the metabolic phenotypes we see using the two inhibitors (Figure 5g-l).

6. Lines 261-262: "generating ATP necessary for lipolysis." This isn't true in any meaningful way. Certainly within the adipocyte, lipolysis is a hydrolysis reaction. The fact that this is wrong causes problems for the rest of the authors' argument in this paragraph.

Response: Thanks for helping us to clarify this, we have re-written this paragraph and removed this statement.

Minor points:

7. That ADIPINT appears to bind PC protein is central to this study. However, this is suddenly introduced on lines 129-130. Out of the handful of potential interactions identified, how did the authors pick this protein? Did they screen all of the hits from supplementary table 11 with pulldowns, and this was the only one that clearly bound ADIPINT (if so, please show this in the supplement)? Or did they pick it simply based on what is known already about PC function?

Response: Comparing the abundance values for each protein in the ADIPINT targeting probe sets 1 and 2 with pulldown samples using the ADIPINT targeting probe sets in the presence of RNase, we found PC to be the only significantly enriched protein detected in this study. We therefore focused our subsequent studies on the ADIPINT-PC interaction. We have added an explanation of how PC was selected to the TROOPS results and methods section.

8. Line 149- the reference to figure 2e is clearly not right.

Response: Here we were referencing that we saw no change in PC protein expression in the proteomic studies and so it is not shown in Fig 2e. We have changed this reference to the proteomic protein expression data given in Supplementary Table 9 and added the expression of PC in all three ASOs for the reader to be able to examine.

9. Take care with interpreting the activity data- while it is great that the authors see differences, suggesting that the ADIPINT-PC interaction is real, such small differences in specific activity are unlikely to drive a meaningful change in substrate flux in vivo.

Response: Thanks for helping us to interpret this data, we have now added in the discussion that the small changes maybe a secondary measure of an altered localization and interactome of the protein. See also the response to point 3 above.

10. How were intracellular oxaloacetate and pyruvate concentrations determined (fig 4d-e)? This needs to be described in detail- the assays for these intracellular metabolites are difficult and easily confounded.

Response: We apologize for overlooking this in the methods section. Oxaloacetate and pyruvate were measured using the commercially available Sigma-Aldrich Kits Cat No. MAK070 and MAK071 respectively. We have added how we carried out these measurements to the methods section.

11. The authors should probably decrease their discussion of the "cytosolic" PC fraction, as this seems like a distraction- PC activity is clearly localized to the mitochondria, and the authors don't need to really understand what the cytosolic band is in order to tell us about the changes in the mitochondrial PC.

Response: We have removed the discussion on the cytosolic PC.

12. Supplementary table 13 should be divided into the two groups that were compared (we should be able to compare the obese and non-obese subjects).

Response: We have now added a new clinical table for both cohort 1 cohort 2 and the newly added cohort 3. Cohorts 2 and 3 are divided into lean/non-obese and obese subjects and are presented in Supplementary Table 14.

13. Figure 6 needs to be interpreted with caution.

The differences in 6c, d, f, and h look as if they would disappear without a single (the highest) point. HOMA-IR correlates with hepatic insulin resistance- which correlates with fatty liver, which increases with obesity. If the authors want to say that there is an effect on adipose insulin resistance, they should try another measure (like Adipo-IR, explained well in PMID: 28465607).

Response: To strengthen the correlation data between ADIPINT expression and the clinical parameters, we have added an additional 20 non-obese and 20 obese women, termed cohort 3. In addition, we thank the author for the suggestion to examine Adipo-IR which we see is strongly correlated with ADIPINT expression and has been added to Figure 6j.

14. Line 209: "These data support the notion that ADIPINT is a physiological regulator of lipid storage and fat cell size". This statement is extremely speculative- if the authors really want to look at the relationship between ADIPINT and physiological lipid storage, that would be a whole new study in itself (perhaps looking at ADIPINT expression in equally obese individuals and comparing this with lipid inappropriately stored in non-adipose tissues).

Response: We have now corrected this statement to keep our conclusions in line with the data presented.

15. Throughout the manuscript the authors talk about a "rate of lipid metabolism", as if "lipid metabolism" was a single entity. This prevents a clear explanation of what is actually happening. Lipolysis, esterification / re-esterification, lipogenesis, and oxidation are all separate processes.

Response: We have now corrected all points where rate of lipid metabolism is mentioned to better reflect the specific metabolic process we are referring to.

16. The English language could use some work. The authors haven't paid attention to some of the basics of English grammar, including very simple concepts like number and tense. While tense matching within a paragraph is often - usually - ignored in scientific writing, it still deserves attention within a sentence.

Response: We have now improved the tense matching within sentences and hope the English has been improved.

Reviewer #3 (Remarks to the Author):

Kerr and authors present a nice human and in vitro adipocyte cell biology study of a human adipocyte specific long non coding RNA, ADIPINT. They show that ADIPINT regulates pyruvate carboxylase (PC) in the mitochondria of human white adipocytes, an enzyme that plays a role in glycolysis and lipogenesis. Knockdown (KD) of ADIPINT reduced pyruvate carboxylase activity and its interactome (but not its protein level) and this corresponded with reductions in intracellular triglycerides and glycerol release from adipocytes, indicating perturbations in lipid metabolism. They also show that KD of ADIPINT reduced oxygen consumption rate and extracellular acidification rate using Seahorse indicating reduction in glycolysis. The authors present focused human translational data on ADIPINT. They show that ADIPINT is increased in the subcutaneous adipose tissue of obese women compared to lean women and corresponding increase in PC activity. ADIPINT expression correlated positively with

adipocyte cell volume, plasma triglycerides, insulin and HOMA-IR which provides correlative support for ADIPINT in regulation of adipocyte lipid homeostasis and lipogenesis. A strength of the paper was identification of an interaction of PC with ADIPINT through a Targeted RNA-protein orthogonal organic phase separation (TROOPS), targeting a specific lncRNA within the after isolating the entire RNA-protein bound proteome. The TROOPS method seems like a valuable improvement on current RNA pull down methods. Overall, the authors present a potentially important finding relating to novel regulatory pathways in adipocytes and energy metabolism, especially in human – yet there are major limitations in its current form that need to be addressed

Overall, this study presents potentially important findings of this lncRNAs ADIPINT role in energy metabolism and also highlights that lncRNAs can have interactions with enzymes such pyruvate carboxylase, thus further expanding our understanding of lncRNA functions, particularly in adipocytes and energy metabolism. However, there are major deficiencies that should be addressed.

Response: We thank the reviewer for the valuable criticism and suggestions for our manuscript.

Major concerns:

1) There are several technical deficiencies in the lncRNA studies. The authors state that ADIPINT transcripts do not have protein coding potential according to CPAT, Rfam, phyloCSF and sORF ribosome coding prediction softwares. This should be validated experimentally e.g., by in vitro translation assay. Although CAGE-seq data is very useful, 5' or 3' RACE experiments should be used to confirm the full length and diversity of isoforms in human adipocytes. There is only one ADIPINT knockdown strategy (ASOs) and no complementary gain of function genetic studies. A second knockdown method should be used to confirm some of the findings? And gain of function genetic studies should be used to complement the knock-down loss of function phenotypes.

Response: We performed PacBio-sequencing in isolated mature adipocytes to identify the ADIPINT full-length transcripts. We find a single isoform that matches the RNA-sequencing signal we see in our human adipocyte cell line. We have added this data to Figure 1c. We have confirmed the effects of ADIPINT knockdown on glycerol release, triglyceride content and lipid synthesis with siRNA and added this to Extended Data Figure 2a-e.

Unfortunately, it has to date, not been possible for us to overexpress any gene in the in vitro differentiated adipocytes using plasmid transfection or virus. We are working on developing such a system to be able to overexpress a given gene but this effort will take a considerable amount of time. However, we believe our addition of a second knockdown strategy provides enough validation of the effect of ADIPINT expression on the mature adipocyte. The siRNA knockdown gave the same results as ASO knockdown, lowering basal lipolysis, triglyceride content and lipid synthesis while not affecting the stimulated lipolysis response.

2) The pyruvate carboxylase (PC) inhibitor studies support the authors hypothesis but are far from conclusive. Only pharmacological approaches were used and no genetic loss-of-function or gain-of-function strategies were applied. Further, there were no significant effects at 5um or 50um, only at 500um, concentrations of oxalate – how specific is this high-concentration? These data need to be validated with genetic strategies. Furthermore, genetics and/or pharmacological data are needed to prove that that ADIPINT acts via PC in mediating effects on adipocyte lipid metabolism - specifically, to prove the mechanistic dependency e.g., through joint manipulations or rescue approaches. Thus, after knock down of PC, was there any effect of ADIPINT KD beyond that of PC? Could overexpression of ADIPINT rescues partial knockdown of PC?

Response: Firstly, we have added a second inhibitor of PC, Avidin (Figure 5d-f). With Avidin treatment we see a dose dependent decrease in glycerol release, triglyceride content and lipid synthesis. We have also knocked down PC using siRNA and see the same effects on lipid breakdown and synthesis as for PC inhibition and ADIPINT knockdown (Figure 5g-k). To demonstrate that ADIPINT acts through PC, we have treated ADIPINT knockdown or control cells with Avidin (Figure 5l). We see no further reduction in triglyceride content with ADIPINT knockdown after PC inhibition. This strongly argues for PC and ADIPINT acting through the same cellular pathway in the fat cells.

3) Furthermore, there is little evidence in this paper or in published work for a causal role of PC in adipocyte/adipose lipid metabolism and pathophysiology in vivo. In the absence of straightforward traditional in vivo mouse modeling of ADIPINT (because of lack of conservation), it would be appropriate to perform adipose-targeted genetic studies of PC in mouse in order to validate the relevance and potential role of PC in adipose pathophysiology in vivo. This is particularly important given the lack of work on the PC pathway in mouse models of obesity, fat distribution, depot storage, and insulin sensitivity etc – i.e., regarding key conclusions about PC and its regulation in human adipocytes by ADIPINT drawn by the authors based on adipocyte studies and human correlative human data.

Response: We kindly point the reviewer to this paper (reference 53 in original MS, PMID 23423574) which shows knockdown of PC in rat liver and adipose tissue results in decreases in body weight, plasma lipid concentrations and profound alterations in adipose glycerol synthesis and fatty acid re-esterification. In addition, it has recently been shown in mice that hepatocyte specific knockdown of PC does not alter body weight or adiposity on chow or high fat diets (PMID 31006591). We have extended our discussion of these studies in the manuscript. Whilst we agree with the reviewer that an adipose specific knockout of PC is still needed to truly confirm the role of PC on adipose pathophysiology, we believe there is evidence of its role in vivo. The focus of this manuscript is how ADIPINT

influences PC function. As ADIPINT does not exist in mouse, knockout models will not illuminate this relationship any further.

4) What is the authors justification for only looking at female obese patients for this study? The lack of males in the discovery and validation human studies needs to be addressed. Further, although the design of human discovery studies is elegant and rigorous, the follow up studies in human are limited and somewhat confusing. Were the samples/data used in the follow up studies the same participants used in the discovery? If so, these are not independent and additional validation and clinical extension should be performed especially as noted above, studies that also include males, and ideally larger numbers than nine individuals.

Response: We have now addressed the female only selection in our clinical cohorts as written above in response to comment 1 from reviewer 1. We have added detail to the methods and results section regarding the follow up cohort which is a completely independent cohort with no individuals used from the original discovery cohort. We have added an additional 20 non-obese and 20 obese women for the correlation analyses in Figure 6e-j. We now stress that we have in three independent cohorts observed a clear relationship between body weight status and ADIPINT and strong correlations between ADIPINT and several clinical variables in a large number of women. Although we cannot say anything about ADIPINT in men, it appears that ADIPINT is linked to the metabolic status in women.

5) Given the proposed role of ADIPINT in mitochondria, it is appropriate to perform some studies of ADIPINT in brown/beige/brite adipocytes – especially as noted in the context of lack of causal mechanistic studies in vivo.

Response: In white fat cells, mitochondria are involved in the regulation of storage/release of lipids and carbohydrate metabolism. In brown fat cells they regulate oxidation of lipid while the role of mitochondria is not well known in beige fat cells. By given the known differences in the function of mitochondria between white and browns adipocytes, we think performing the additional experiments in brown adipocytes will not be able to provide further causal mechanistic insight into the function of ADIPINT in regulating the storage/release of lipids or carbohydrate metabolism in obesity in vivo. We discover ADIPINT to be regulated by obesity in white fat cells and consequently focus on important white cell functions altered during obesity. With the lack of commercially available brown/beige human fat cells it is not feasible within the time frame (6 months) of this revision to establish a brown adipocyte cell line to perform such studies.

6) The conclusions in discussion should be toned down. This work is a far distance from providing evidence that “ADIPINT-PC interactions could have an important clinical role and constitute a pharmacological target for metabolic diseases.”.

Response: We agree with the reviewer the conclusions were too strong and we have toned down such statements in the discussion.

Minor

1) Line 77: “Transcript 2 is 4.8 kb long and contains two exons (Fig. 1c); the junction was validated by qPCR and Sanger sequencing (data not shown).” The authors should show this qPCR data and a BLAT map of the amplicon that was sent for Sanger sequencing in supplement?

Response: We have now added the PacBio-sequencing demonstrating the full length transcripts matching transcript 2 (Figure 1c).

2) Supplement table 2 should contain the genomic location of the lincRNAs?

Response: We now add the genomic location of the lincRNAs differentially expressed following bariatric surgery.

3) The FISH experiment Figure 1h. Can the authors include a merged image of HOECHST and ADIPINT only, as the BODIPY make it difficult to see the sub cellular location?

Response: We have now added a merged image of HOESCHT and ADIPINT to Figure 1h.

4) Did the Gapmer Knockdown effect differentiation of adipocytes (Figure 2)?

Response: To minimize any effect on differentiation we always knocked down ADIPINT in the mature adipocyte. Knockdown of ADIPINT did decrease several adipocyte marker genes and reduced the lipid content of the cells and so it could be considered to alter the differentiation of the adipocyte. However, as many important adipocyte genes such

as PLIN1, ADIPOQ and FABP4 were not altered, we conclude the effect on differentiation may be secondary to the effect on PC.

5) Line 149: 'PC expression was not altered at the protein level after ADIPINT knockdown (Fig 2e)' – There is no protein expression of PC in Fig2e. Please correct this and include in figures.

Response: We now point readers to the proteomic data in in Supplementary Table 9 as addressed above.

6) The n numbers for each experiment should be indicated in each figure legend.

Response: Each individual n is plotted in the figures and if not it is given in the figure legend.

7) Line 160: Typo "thetotal" needs a space 'the total'

Response: Now corrected.

8) Line 207: Typo "fat cell volume and ADIPINT expression (Fig. 6i)" – Should be Fig.6e as displayed

Response: Now corrected.

REVIEWER COMMENTS

Reviewer #1 (Remarks to the Author):

REVIEWER COMMENTS Reviewer #1 (Remarks to the Author): The manuscript “The long noncoding RNA ADIPINT is a gatekeeper of pyruvate carboxylase function regulating human fat cell metabolism” examines an lncRNA ADIPINT as a regulator of lipid metabolism, by binding and regulating pyruvate carboxylase enzyme in mitochondria. ADIPINT knockdown via RNA interference lowers metabolic activity according to several measures. A novel pulldown strategy to examine ADIPINT protein binding partners identified pyruvate carboxylase. ADIPINT also appears to encourage mitochondrial protein binding of pyruvate carboxylase to other proteins involved in glycolysis. Overall, the manuscript is very well written, and they carried out right experiments to support their hypothesis. While I am supportive of this manuscript, a few additional clarifications listed below is warranted.

A. Previous concerns:

1. Page 6, line 143-144 "RNA-FISH with immunofluorescence staining for ADIPINT and PC in tandem confirmed the localization overlap (Fig. 3h). Together these data suggest a physical ADIPINT-PC interaction". I cannot see this overlap in the immunofluorescence images.

Response: To better demonstrate the overlap in fluorescence signal for ADIPINT and PC, we have revised the presentation of the RNA-FISH images. We also added profile plots of pixel intensity for ADIPINT and PC at selected regions. Data are now in Figure 3m.

Reviewer: I still find the speckled distribution of ADIPINT to be unconvincing, from an interaction standpoint. The utility of the profile plot of pixel intensity is unclear, peaks don't match and the three spots selected appear to be chosen for no specific reason. At best, this is weak co-localization.

2) The pyruvate carboxylase (PC) inhibitor studies support the authors hypothesis but are far from conclusive. Only pharmacological approaches were used and no genetic loss-of-function or gain-of-function strategies were applied. Further, there were no significant effects at 5um or 50um, only at 500um, concentrations of oxalate – how specific is this high-concentration? These data need to be validated with genetic strategies. Furthermore, genetics and/or pharmacological data are needed to prove that that ADIPINT acts via PC in mediating effects on adipocyte lipid metabolism - specifically, to prove the mechanistic dependency e.g., through joint manipulations or rescue approaches. Thus, after knock down of PC, was there any effect of ADIPINT KD beyond that of PC? Could overexpression of ADIPINT rescues partial knockdown of PC?

Response: Firstly, we have added a second inhibitor of PC, Avidin (Figure 5d-f). With Avidin treatment we see a dose dependent decrease in glycerol release, triglyceride content and lipid synthesis. We have also knocked down PC using siRNA and see the same effects on lipid breakdown and synthesis as for PC inhibition and ADIPINT knockdown (Figure 5g-k). To demonstrate that ADIPINT acts through PC, we have treated ADIPINT knockdown or control cells with Avidin (Figure 5l). We see no further reduction in triglyceride content with ADIPINT knockdown after PC inhibition. This strongly argues for PC and ADIPINT acting through the same cellular pathway in the fat cells.

Reviewer: This new data is good, but a rescue experiment would be better.

3) Furthermore, there is little evidence in this paper or in published work for a causal role of PC in adipocyte/adipose lipid metabolism and pathophysiology in vivo. In the absence of straightforward traditional in vivo mouse modeling of ADIPINT (because of lack of conservation), it would be appropriate to perform adipose-targeted genetic studies of PC in mouse in order to validate the relevance and potential role of PC in adipose pathophysiology in vivo. This is particularly important given the lack of work on the PC pathway in mouse models of obesity, fat distribution, depot storage, and insulin sensitivity etc – i.e., regarding key conclusions about PC and its regulation in human adipocytes by ADIPINT drawn by the authors based on adipocyte studies and human correlative human data.

Response: We kindly point the reviewer to this paper (reference 53 in original MS, PMID 23423574) which shows knockdown of PC in rat liver and adipose tissue results in decreases in body weight, plasma lipid concentrations and profound alterations in adipose glycerol synthesis and fatty acid re-esterification. In addition, it has recently been shown in mice that hepatocyte specific knockdown of PC does not alter body weight or adiposity on chow or high fat diets (PMID 31006591). We have extended our discussion of these studies in the manuscript. Whilst we agree with the reviewer that an adipose specific knockout of PC is still needed to truly confirm the role of PC on adipose pathophysiology, we believe there is evidence of its role in vivo. The focus of this manuscript is how ADIPINT influences PC function. As ADIPINT does not exist in mouse, knockout models will not illuminate this relationship any further.

Reviewer: This comment strikes me as beyond the current paper's scope, which is on ADIPINT's effects on PC.

B. Additional new concerns for authors to look into:

- Extended Data Figure 3 I: PC is a mitochondrial protein (<https://www.uniprot.org/uniprot/P11498>). Why is there a signal in the cytoplasmic fraction?! It should just be mitochondrial, like TOM20. I think the authors should add a positive control for the cytoplasmic fraction, like actin or tubulin, to see the quality of their cellular fractionation, or test the PC antibody because the signal should be just in the mitochondria.

- Extended Data Figure 3 N: Authors note "RNA-FISH with immunofluorescence staining for ADIPINT and PC in tandem confirmed the localization overlap (Fig. 3m). Interestingly, ADIPINT signal only partially overlapped with TOM20 (Extended Data Fig. 3n). Together these data suggest a physical ADIPINT-PC interaction." Since PC and TOM20 are both strictly mitochondrial proteins, the immunofluorescence signal of these 2 proteins should overlap. How is it possible that ADIPINT does not co-localize with TOM20? Moreover, since ADIPINT is just a few dots sparse all over the cell (from the figure 3m ADIPINT also localizes in nuclei), I don't think it is possible to conclude its cellular localization with a simple immunofluorescence experiment. Especially since they don't see an overlap between ADIPINT and TOM20. I think there is something wrong. Since PC protein was found even in the cytoplasmic fraction (Extended Data Figure 3 I) maybe the antibody is not specific?! Even the fact that they see TOM20 and GLS (2 strictly mitochondrial proteins) in the nuclear fraction (Figure 4 F) suggests that their cellular fractionation protocol needs to be improved.

Reviewer #2 (Remarks to the Author):

I very much appreciate the amount of work the authors put into this revision. My major concerns have been dealt with. Furthermore, I feel the discussion in particular is far more clear and consistent with the results in this draft than in the prior draft.

A few items should be further addressed:

2a) This explanation of normalization doesn't make sense. The authors need to explain their normalization approach better. If all normalization occurred within-experiment against average of these variables, differences (or lack of differences) could be arbitrary. I am sure this is not what the investigators meant.

2c) Thank you for plotting the Seahorse data in absolute terms! This makes it abundantly clear that the difference between the groups lies in basal respiration- a conclusion that couldn't have been easily derived from the transformed data in the main text. For the sake of investigators who routinely read Seahorse data, Extended Data Figures 2g-h need to be moved out of the supplement into the main figures.

Also, as the authors didn't perform a mitochondrial stress test, it will be clearer to those who routinely work in this area to call the data they are calling "ATP-linked OCR" something that reflects the actual experiment that was performed. Perhaps "Glycolytic Capacity".

3) I don't believe an in vitro experiment with the synthetic ADIPINT is necessary. However, I don't understand why the authors think the synthetic ADIPINT wouldn't affect the interactome in this experiment, per their model.

12) The cohort 2 HOMA-IR is mathematically impossible. If glucose is ~10% higher, and insulin is ~3x higher, HOMA-IR cannot be cut in half. The authors need to re-do this calculation.

Reviewer #3 (Remarks to the Author):

The authors have responded satisfactorily to the concerns raised. There remain a couple of minor concerns.

1. In response to "Did the Gapmer Knockdown effect differentiation of adipocytes (Figure 2)", the authors provided the response below but I do not see where the data associated with this response is presented.

2. In the abstract the authors state "ADIPINT knockdown alters the interactome and decreases the mitochondrial abundance" I don't see where mitochondrial mass has been measured but they do measure mitochondrial activity. Please correct or highlight abundance/mass measurement.

REVIEWER COMMENTS

Reviewer #1 (Remarks to the Author):

REVIEWER COMMENTS Reviewer #1 (Remarks to the Author): The manuscript "The long noncoding RNA ADIPINT is a gatekeeper of pyruvate carboxylase function regulating human fat cell metabolism" examines an lncRNA ADIPINT as a regulator of lipid metabolism, by binding and regulating pyruvate carboxylase enzyme in mitochondria. ADIPINT knockdown via RNA interference lowers metabolic activity according to several measures. A novel pulldown strategy to examine ADIPINT protein binding partners identified pyruvate carboxylase. ADIPINT also appears to encourage mitochondrial protein binding of pyruvate carboxylase to other proteins involved in glycolysis. Overall, the manuscript is very well written, and they carried out right experiments to support their hypothesis. While I am supportive of this manuscript, a few additional clarifications listed below is warranted.

Response: We thank the reviewer for supporting this manuscript and hope we can address the remaining concerns.

A. Previous concerns:

1. Page 6, line 143-144 "RNA-FISH with immunofluorescence staining for ADIPINT and PC in tandem confirmed the localization overlap (Fig. 3h). Together these data suggest a physical ADIPINT-PC interaction". I cannot see this overlap in the immunofluorescence images.

Response: To better demonstrate the overlap in fluorescence signal for ADIPINT and PC, we have revised the presentation of the RNA-FISH images. We also added profile plots of pixel intensity for ADIPINT and PC at selected regions. Data are now in Figure 3m.

Reviewer: I still find the speckled distribution of ADIPINT to be unconvincing, from an interaction standpoint. The utility of the profile plot of pixel intensity is unclear, peaks don't match and the three spots selected appear to be chosen for no specific reason. At best, this is weak co-localization.

Response: LncRNA is often expressed at a lower level relative to their interacting protein, with lncRNA expressed 0.3-1000 copies per cell and proteins at 10000-80000000 copies per cell (PMID: 34526358). The speckled distribution of ADIPINT reflects an abundance within the range for a lncRNA and we estimate ADIPINT to be expressed at 10-60 copies per cell with RNA-FISH. Previous publications have demonstrated that lncRNA with similar speckled expression are able to regulate their interacting protein(s) in a meaningful way (PMID: 34879239, 32259487, 32440655). Although the imaging data does not demonstrate an exclusive overlap, we point to the fact we have used three independent methods in addition to the IF images to confirm the interaction, including electron microscopy. All four methods point towards an interaction and is strong combined evidence. We now add to the discussion the following statement.

'However, it needs to be investigated how the lower expressed ADIPINT can regulate the highly abundant enzyme, PC. This obstacle may demand development of novel methods to precisely determine the stoichiometry between ADIPINT and PC in specific subcellular compartments. A similar relationship between a lncRNA and interacting metabolic enzymes was recently seen in breast cancer cells (PMID: 34879239).'

2) The pyruvate carboxylase (PC) inhibitor studies support the authors hypothesis but are far from

conclusive. Only pharmacological approaches were used and no genetic loss-of-function or gain-of-function strategies were applied. Further, there were no significant effects at 5um or 50um, only at 500um, concentrations of oxalate – how specific is this high-concentration? These data need to be validated with genetic strategies. Furthermore, genetics and/or pharmacological data are needed to prove that ADIPINT acts via PC in mediating effects on adipocyte lipid metabolism - specifically, to prove the mechanistic dependency e.g., through joint manipulations or rescue approaches. Thus, after knock down of PC, was there any effect of ADIPINT KD beyond that of PC? Could overexpression of ADIPINT rescue partial knockdown of PC?

Response: Firstly, we have added a second inhibitor of PC, Avidin (Figure 5d-f). With Avidin treatment we see a dose dependent decrease in glycerol release, triglyceride content and lipid synthesis. We have also knocked down PC using siRNA and see the same effects on lipid breakdown and synthesis as for PC inhibition and ADIPINT knockdown (Figure 5g-k). To demonstrate that ADIPINT acts through PC, we have treated ADIPINT knockdown or control cells with Avidin (Figure 5l). We see no further reduction in triglyceride content with ADIPINT knockdown after PC inhibition. This strongly argues for PC and ADIPINT acting through the same cellular pathway in the fat cells.

Reviewer: This new data is good, but a rescue experiment would be better.

Response: We agree with the reviewer that a rescue experiment would provide further evidence for the ADIPINT-PC effect on lipid metabolism. We unfortunately have been unable to overexpress any gene in our system to date and so have not been able to perform the rescue experiment. We address this in a previous comment (Reviewer 3 Major concern 1) in the original rebuttal letter and list it below.

'Unfortunately, it has to date, not been possible for us to overexpress any gene in the in vitro differentiated adipocytes using plasmid transfection or virus. We are working on developing such a system to be able to overexpress a given gene but this effort will take a considerable amount of time'.

3) Furthermore, there is little evidence in this paper or in published work for a causal role of PC in adipocyte/adipose lipid metabolism and pathophysiology in vivo. In the absence of straightforward traditional in vivo mouse modeling of ADIPINT (because of lack of conservation), it would be appropriate to perform adipose-targeted genetic studies of PC in mouse in order to validate the relevance and potential role of PC in adipose pathophysiology in vivo. This is particularly important given the lack of work on the PC pathway in mouse models of obesity, fat distribution, depot storage, and insulin sensitivity etc – i.e., regarding key conclusions about PC and its regulation in human adipocytes by ADIPINT drawn by the authors based on adipocyte studies and human correlative human data.

Response: We kindly point the reviewer to this paper (reference 53 in original MS, PMID 23423574) which shows knockdown of PC in rat liver and adipose tissue results in decreases in body weight, plasma lipid concentrations and profound alterations in adipose glycerol synthesis and fatty acid re-esterification. In addition, it has recently been shown in mice that hepatocyte specific knockdown of PC does not alter body weight or adiposity on chow or high fat diets (PMID 31006591). We have extended our discussion of these studies in the manuscript. Whilst we agree with the reviewer that an adipose specific knockout of PC is still needed to truly confirm the role of PC on adipose pathophysiology, we believe there is evidence of its role in vivo. The focus of this manuscript is how ADIPINT influences PC function. As ADIPINT does not exist in mouse, knockout models will not illuminate this relationship any further.

Reviewer: This comment strikes me as beyond the current paper's scope, which is on ADIPINT's effects on PC.

Response: We agree with the review that the causal role of PC in adipocyte/adipose lipid metabolism and pathophysiology will need to be further fully investigated. However, in the submitted revisions, we have added the data with loss – function experiment on PC to demonstrate the important role of the PC at the regulation of lipid metabolism in human adipocytes. We feel all these data provide strong support for the importance of the ADIPINT-PC axis in the regulation of lipid metabolism in human adipocyte. The data are also added to answer previous concerns raised by the other reviewers.

B. Additional new concerns for authors to look into:

- Extended Data Figure 3 I: PC is a mitochondrial protein (<https://www.uniprot.org/uniprot/P11498>). Why is there a signal in the cytoplasmic fraction?! It should just be mitochondrial, like TOM20. I think the authors should add a positive control for the cytoplasmic fraction, like actin or tubulin, to see the quality of their cellular fractionation, or test the PC antibody because the signal should be just in the mitochondria.

Response: To validate our fractionation protocol in this study, we have now added GAPDH as a positive control for the cytoplasmic fraction (Extended Data Fig. 3I) where it is enriched. We validate the PC antibody through siRNA knockdown of PC shown in Fig. 5h. In addition, we detected PC using mass-spectrometry in the cytoplasmic fraction and found it to be highly abundant (Extended Data Fig. 4e). We could enrich PC ~12 fold after immunoprecipitation in the cytoplasmic fraction using the PC primary antibody followed by mass-spectrometry. Furthermore, we detect the cytosolic band of PC with two separate primary antibodies against PC (Cat No. PA5-50101 in manuscript, data not shown for antibody Cat No. 16588-1-AP in fractionation experiments). We do not detect TOM20 or Glutaminase in the cytoplasmic fraction and so do not believe the PC expression to be the result of mitochondrial contamination.

- Extended Data Figure 3 N: Authors note "RNA-FISH with immunofluorescence staining for ADIPINT and PC in tandem confirmed the localization overlap (Fig. 3m). Interestingly, ADIPINT signal only partially overlapped with TOM20 (Extended Data Fig. 3n). Together these data suggest a physical ADIPINT-PC interaction." Since PC and TOM20 are both strictly mitochondrial proteins, the immunofluorescence signal of these 2 proteins should overlap. How is it possible that ADIPINT does not co-localize with TOM20? Moreover, since ADIPINT is just a few dots sparse all over the cell (from the figure 3m ADIPINT also localizes in nuclei), I don't think it is possible to conclude its cellular localization with a simple immunofluorescence experiment. Especially since they don't see an overlap between ADIPINT and TOM20. I think there is something wrong. Since PC protein was found even in the cytoplasmic fraction (Extended Data Figure 3 I) maybe the antibody is not specific?! Even the fact that they see TOM20 and GLS (2 strictly mitochondrial proteins) in the nuclear fraction (Figure 4 F) suggests that their cellular fractionation protocol needs to be improved.

Response: To further validate the cellular fractionation in Fig. 4f we have added GAPDH. Mitochondrial proteins have been detected in nuclear fractions in other cell types (PMID:26520802, 28086092) and we do not detect PC or GAPDH in the nuclear fraction. As addressed above, we believe the expression of PC to not be solely limited to the mitochondria in human adipocytes. PC and TOM20 demonstrate different localization profiles within the adipocyte, although they share a large overlap (Extended Data Fig 3I). As PC and TOM20 have different localization profiles it is quite possible that

ADIPINT co-localizes with PC and not the strictly mitochondrial protein TOM20. In the re-revised MS we discuss this better.

Reviewer #2 (Remarks to the Author):

I very much appreciate the amount of work the authors put into this revision. My major concerns have been dealt with. Furthermore, I feel the discussion in particular is far more clear and consistent with the results in this draft than in the prior draft.

Response: We thank the reviewer for the positive comments and are glad all their major concerns have been dealt with.

A few items should be further addressed:

2a) This explanation of normalization doesn't make sense. The authors need to explain their normalization approach better. If all normalization occurred within-experiment against average of these variables, differences (or lack of differences) could be arbitrary. I am sure this is not what the investigators meant.

Response: We agree that our explanation of how the data was normalized was not clear and provide a more detailed explanation here and to the methods section. The triglyceride/glycerol was normalized by the amount of protein for each well in an experiment first. To standardize the experiments (and make them comparable), the value from the first step was normalized by the mean. The normalization we use here is a standard way to control for variations in the total amount of triglyceride/glycerol after the human pre-adipocytes have differentiated into mature adipocytes. By normalizing each value by the average amount of triglyceride/glycerol present in the cells for that particular differentiation, the analysis examines the consistent directional and fold difference after ADIPINT knockdown. We hope this clarifies the methodology to the reviewer.

2c) Thank you for plotting the Seahorse data in absolute terms! This makes it abundantly clear that the difference between the groups lies in basal respiration- a conclusion that couldn't have been easily derived from the transformed data in the main text. For the sake of investigators who routinely read Seahorse data, Extended Data Figures 2g-h need to be moved out of the supplement into the main figures.

Also, as the authors didn't perform a mitochondrial stress test, it will be clearer to those who routinely work in this area to call the data they are calling "ATP-linked OCR" something that reflects the actual experiment that was performed. Perhaps "Glycolytic Capacity".

Response: We have now moved Extended Data Fig. 2g-h to the main figure and renamed 'ATP-linked OCR' to 'Glycolytic Capacity'.

3) I don't believe an in vitro experiment with the synthetic ADIPINT is necessary. However, I don't understand why the authors think the synthetic ADIPINT wouldn't affect the interactome in this experiment, per their model.

Response: Thank you for giving us the chance to clarify. We meant that as we are measuring the PC activity after lysing of the cell and sonication, the artificial environment may not allow ADIPINT to first restore the complete interactome of PC (which may be through regulating its cellular localization) and then secondly alter the activity. If there was or was not an effect after spiking ADIPINT into the cell lysate may just confirm if the intact cell architecture is necessary for function of the lncRNA.

12) The cohort 2 HOMA-IR is mathematically impossible. If glucose is ~10% higher, and insulin is ~3x higher, HOMA-IR cannot be cut in half. The authors need to re-do this calculation.

Response: We thank the reviewer for pointing out this error, we have now checked and corrected the HOMA-IR for Cohort 2 in the clinical table.

Reviewer #3 (Remarks to the Author):

The authors have responded satisfactorily to the concerns raised. There remain a couple of minor concerns.

Response: We thank the reviewer for the positive response and address the remaining concerns

1. In response to "Did the Gapmer Knockdown effect differentiation of adipocytes (Figure 2)", the authors provided the response below but I do not see where the data associated with this response is presented.

Response: Thank you for pointing this out, we have added the expression of PLIN1, FABP4 and ADIPOQ from the microarray for all three GapmeRs and control into Supplementary Table 5.

2. In the abstract the authors state "ADIPINT knockdown alters the interactome and decreases the mitochondrial abundance" I don't see where mitochondrial mass has been measured but they do measure mitochondrial activity. Please correct or highlight abundance/mass measurement.

Response: Here we mean that ADIPINT decreased the mitochondrial abundance of PC not the total mitochondrial mass of the adipocytes. The new sentence reads 'ADIPINT knockdown alters the interactome and decreases the abundance and enzymatic activity of PC in mitochondria.'

REVIEWERS' COMMENTS

Reviewer #1 (Remarks to the Author):

Authors have indeed responded to my previous concerns adequately. Although, I'm still not convinced about the immunofluorescence experiments (based on the previous concern), authors appear to argue that they have validated the interaction through orthogonal methods and hence they are confident that ADIPINT interacts with PC. Since immunofluorescence is not informative in this context, I suggest to remove it altogether without affecting the quality of the work that went into this manuscript. Otherwise, I am largely supportive of this manuscript to getting published.

Reviewer #2 (Remarks to the Author):

I appreciate the additional clarifications. My concerns have been addressed.

Reviewer #1 (Remarks to the Author):

Authors have indeed responded to my previous concerns adequately. Although, I'm still not convinced about the immunofluorescence experiments (based on the previous concern), authors appear to argue that they have validated the interaction through orthogonal methods and hence they are confident that ADIPINT interacts with PC. Since immunofluorescence is not informative in this context, I suggest to remove it altogether without affecting the quality of the work that went into this manuscript. Otherwise, I am largely supportive of this manuscript to getting published.

We have now removed the immunofluorescence experiments as you and the reviewer suggested.

We resubmit a reformatted version of the manuscript and will make all data available. Here is a brief summary of the findings as requested in the author checklist.

The fat cell-specific long non-coding RNA, ADIPINT, regulates the lipid content in human white adipocytes through altering the localization and interacting proteins of the enzyme pyruvate carboxylase.

We would once again like to thank you for the clear and professional help in processing our manuscript you provided and we look forward to working with you again in the future.

Kind Regards

Alastair, Peter and Hui